# Left-right cortical interactions drive intracellular pattern formation in the ciliate *Tetrahymena*

Chinkyu Lee[1☯], Ewa Joachimiak[2☯], Wolfgang Maier[3☯], Yu-Yang Jiang[1], Mireya Parra[1], Karl F. Lechtreck[1], Eric S. Cole[4], Jacek Gaertig[1]*

1 Department of Cellular Biology, University of Georgia, Athens, Georgia, United States of America,
2 Nencki Institute of Experimental Biology of Polish Academy of Sciences, Warsaw, Poland,
3 Bioinformatics, University of Freiburg, Freiburg, Germany, 4 Biology Department, St. Olaf College, Northfield, Minnesota, United States of America

☯ These authors contributed equally to this work.
* jgaertig@uga.edu

## Abstract

In ciliates, cortical organelles, including ciliary arrays, are positioned at precise locations along two polarity axes: anterior-posterior and circumferential (lateral). We explored the poorly understood mechanism of circumferential patterning, which generates left-right asymmetry. The model ciliate *Tetrahymena* has a single anteriorly-located oral apparatus. During cell division, a single new oral apparatus forms near the equator of the parental cell and along the longitude of the parental organelle. Cells homozygous for *hypoangular 1* (*hpo1*) alleles, assemble multiple oral apparatuses positioned either to the left or right flanking the normal oral longitude. Using comparative next-generation sequencing, we identified *HPO1* as a gene encoding an ARMC9-like protein. Hpo1 colocalizes with the ciliary basal bodies, forming a bilateral concentration gradient, with the high point on the cell's right side and a sharp drop-off that marks the longitude at which oral development initiates on the ventral side. A second Hpo1 concentration drop-off is present on the dorsal surface, where it marks the position for development of a cryptic oral apparatus that forms in the *janus* mutants. Hpo1 acts bilaterally to exclude oral development from the cell's right side. Hpo1 interacts with the Beige-Beach domain protein Bcd1, a cell's left side-enriched factor, whose loss also confers multiple oral apparatuses on the ventral surface. A loss of both Hpo1 and Bcd1 is lethal and profoundly disrupts the positioning, organization and size of the forming oral apparatus (including its internal left-right polarity). We conclude that in ciliates, the circumferential patterning involves gradient-forming factors that are concentrated on either the cell's right or left side and that the two sides of the cortex interact to create boundary effects that induce, position and shape developing cortical organelles.

**Data availability statement:** All relevant data are within the paper and its Supporting Information files. The raw NGS data were deposited at the public SRA database (Bioproject PRJNA1216161).

**Funding:** This work was supported by the National Institutes of Health (grant R01GM135444 to J.G., R01GM110413 to K.F.L.), National Science Foundation (grant 1947608 to E.C.) and the German Federal Ministry of Education and Research (Bundesministerium für Bildung und Forschung) (grant 031L0101C de.NBI-epi to W.M.). The funders had no role in study design, data collection and analysis, decision to publish, or preparation of the manuscript.

**Competing interests:** The authors have declared that no competing interests exist.

## Author summary

This study investigates how organelles are organized into precise patterns within the cell. The authors used the multiciliated protist *Tetrahymena* as a model. In this single-celled organism, the oral apparatus, a feeding organelle composed of multiple rows of cilia, is consistently positioned at a single location, along both the anterior-posterior and left-right axes. In the *hypoangular* mutants, the oral apparatus shifts to the left or right of its normal position. This defect arises during cell division, when a new oral apparatus forms in one of the two daughter cells. Through genetic crosses and whole genome sequencing, the authors identified the *hypoangular* gene product as a protein called Hpo1, which resembles the conserved cilia-associated protein ARMC9. Hpo1 localizes to the base of cilia and forms a bidirectional concentration gradient on the right side of the *Tetrahymena* cell. Hpo1 interacts with Bcd1, a protein concentrated on the left side of the cell. Interactions among Hpo1, Bcd1, and another gene product, JanC, determine the precise position of the developing oral apparatus and ensure that only one organelle forms during each generation. In summary, the study reveals an unexpectedly sophisticated pattern formation mechanism that involves gradient-forming proteins and operates within a single cell.

## Introduction

Eukaryotic cells typically have a prominent anterior-posterior (also called front-to-rear or apical-basal) polarity. In addition, some cell types, including cells that move while adhering to a substrate or single-cell metazoan embryos, also have a distinguishable dorsal-ventral polarity axis. Moreover, some cells display chiral asymmetries in reference to the anterior-posterior axis. Ciliated protists have features that are favorable for exploring how multiple polarities and chiral asymmetries are generated in the cell. In the genetic model ciliate *Tetrahymena thermophila,* cortical organelles occupy specific positions along the anterior-posterior axis and around the cell circumference (Fig 1A-1). All cortical structures are duplicated during a type of cell division called "tandem duplication", in the course of which a single parental cell is subdivided into two daughter cells arranged head-to-tail (Fig 1A). The anterior daughter cell assembles new posterior organelles while the posterior daughter cell assembles new anterior organelles. A new oral apparatus (oral primordium) forms on the ventral surface near the future anterior end of the posterior daughter cell (Fig 1A-2). Remarkably, the primordium almost always assembles in association with the basal bodies of a single specific ciliary row (row 0, Fig 1B). The anterior-posterior position of the oral primordium is regulated by a set of conserved cortical proteins (including components of the Hippo signaling pathway) that appear to act by marking cortical domains where organelle assembly is disallowed (reviewed in [2]). The principles of lateral positioning (around the circumferential axis) are less understood, but could be exceptionally revealing as the responsible mechanism must generate precise coordinates, in the

PLOS Genetics

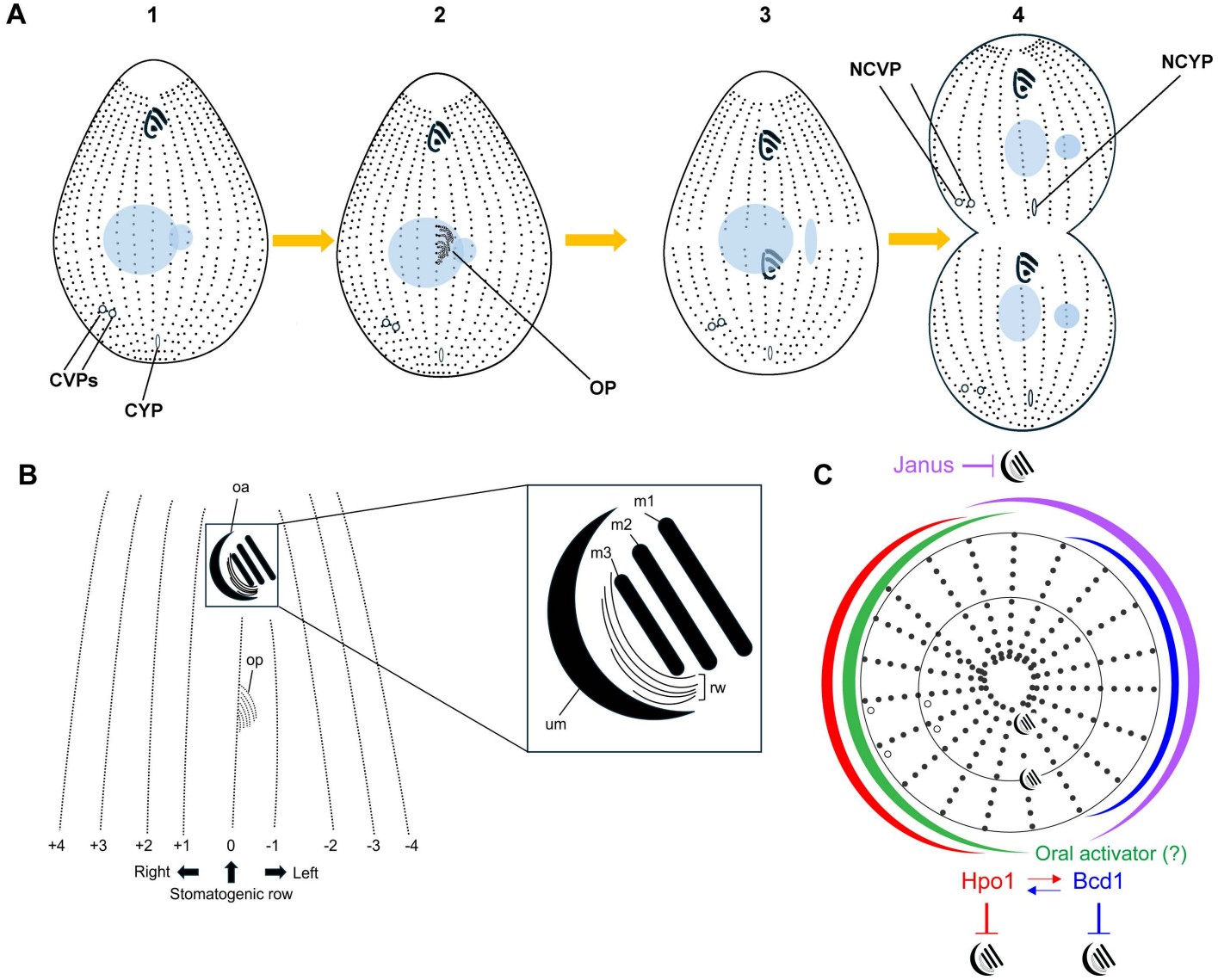

**Fig 1. Circumferential positioning In *Tetrahymena*. (A)** Cell cycle stages of *T. thermophila* with emphasis on cortical development. Note that during cell division, new organelles form at same longitudes (same ciliary rows) as the old organelles. **(B)** A detailed view of the stage 2 cell presents the row numbering method. **(C)** A multi-domain model for circumferential positioning in a ciliate. The Janus activity is proposed to be located in the dorsal region based on the known distribution of one of its components, JanA [1]. Abbreviations: OA, oral apparatus; OP, oral primordium; CVPs, contractile vacuole pores; NCVPs, new contractile vacuole pores; cyp, cytoproct: ncyp, new cytoproct; m1, m2, m3 oral M (membranelle) rows; um, oral undulating membrane row; rw, ribbed wall of microtubules.

absence of structural landmarks available for the anterior-posterior positioning (such as cell ends). Furthermore, studies in ciliates may provide insights into the mechanisms that generate chiral asymmetries. In multicellular models, mutations in cytoskeletal proteins (including myosin, α-tubulin and β-tubulin) can invert the left-right positioning of body organs ([3–6], reviewed in [7]). Intriguingly in ciliates, alterations of left-right organelle positioning can be caused by mutations in non-cytoskeletal proteins (including a protein associated with vesicle trafficking and a kinase [8,1]) or by trauma that alters the cortical organization in a way that is epigenetically heritable across multiple cell generations [9,10].

*Tetrahymena thermophila* is a fast growing ciliate that is widely used as a genetic model with both reverse [11] and forward [12] genetics strategies. A library of unique *Tetrahymena* mutants with defects in cortical patterning has been developed and extensively characterized by Joseph Frankel and colleagues (University of Iowa) [13,14]. The mutants of interest here are those in which the principal defect is in the positioning of organelles around the circumferential axis. The *hpo1* and *bcd1* alleles [8,15,16] exhibit multiple oral primordia assembled at incorrect circumferential locations. Recently, Cole and colleagues identified the *BCD1* gene product as a Beige-BEACH domain protein orthologous to the mammalian NBEA and Rugose of *D. melanogaster*. Bcd1 is enriched on the cell's left side and appears to act by regulating the balance between delivery and retrieval of organelle assembly components through endocytosis and exocytosis, to control the size of cortical domains competent for organelle formation [8]. Here, we used comparative next-generation sequencing (NGS), to identify *HPO1* as a gene encoding a protein similar to ARMC9, a highly conserved ciliary distal tip protein. Remarkably, Hpo1 is enriched on the *Tetrahymena* cell's right side where it forms a bilateral concentration gradient that diminishes both dorsally and ventrally. Oral development occurs at positions at which Hpo1 levels drop abruptly. Our data suggest that Hpo1 functions as a bidirectional repressor excluding oral morphogenesis from the cell's right lateral side. We also reveal that together, the right-side-biased Hpo1 and the left-side-biased Bcd1 are required for cell viability and that mutants lacking both proteins display diverse and severe patterning defects, including abnormalities in the internal organization of cortical structures. We propose that in ciliates, circumferential patterning involves multiple cortical subdomains, formed by bilateral gradients of pattern regulators that interact with each other and act by generating cortical boundaries that define sites of organelle assembly.

## Results

### Hpo1 is an ARMC9-like protein, TTHERM_001276421

*Tetrahymena* cells have two permanent polarity axes: anterior-posterior (AP) and circumferential (C). AP polarity is reflected by an asymmetric placement of major cortical organelles: the oral apparatus (OA) near the anterior cell end, the contractile vacuole pores (CVPs) and the cytoproct near the posterior cell end (Fig 1A-1). There are about 20 longitudinal (locomotory) ciliary rows that circumvent the cell. While most of the longitudinal rows span the entire length of the cell, on the ventral side, two rows (called postoral left and right) are shorter, as they terminate posteriorly to the OA (Fig 1A-1). C axis polarity is revealed by asymmetric lateral positions of cortical organelles. While the cytoproct is located at the same longitude as the OA, the CVPs are located on the cell's right lateral side (Fig 1A-1). During cell division, new organelles form at the same cell longitudes as the preexisting organelles. The new OA (oral primordium or OP) assembles initially as a group of ciliary basal bodies (BBs) forming at a subequatorial AP position in the middle of the cell's ventral side, on the left side of the right postoral row (Fig 1A-2). BBs proliferate and form four oral ciliary rows (UM on the right and M1, M2 and M3 on the left) (Fig 1A-2, 1B). In the advanced stage of oral development, a fission zone bisects all somatic ciliary rows anteriorly to the OP (Fig 1A-3). The new CVPs and the new cytoproct form at the posterior ends of the anterior half-rows (Fig 1A-4). The cell completes nuclear divisions and undergoes cytokinesis (Fig 1A-4).

We adopted the longitudinal row numbering scheme after Cole and colleagues [15]. The right postoral (stomatogenic) row is designated as row 0 (Fig 1B). The remaining rows are numbered as +1 or higher when moving to the cell's right side or -1 or lower when moving to the cell's left side (Fig 1B). *hpo1* (*hypoangular 1*) alleles disturb the positioning of oral development along the C axis [16]. While in the wild type, a single OP forms near the left side of row 0, in the *hpo1* mutants, multiple OPs form near several adjacent ventral rows located to either the left (rows -1, -2…) or right (rows +1,+2…) of the stomatogenic row 0 (Fig 2B, 2E compare to Fig 2A, 2D, 2G, 2H). One complication is that in some *hpo1* mutant cells, the oral apparatus is wider than normal, which is correlated with the presence of 3 post-oral rows. In such cases we designated the middle postoral row as 0 [16]. As described by Frankel and colleagues [16], in the *hpo1-3* mutants, the extra OPs formed more frequently to the right of row 0 (Fig 2B, 2E, 2H). However, when multiple OPs were present, they were located near adjacent rows, typically including row 0 or a row in the proximity of row 0. Thus, in the

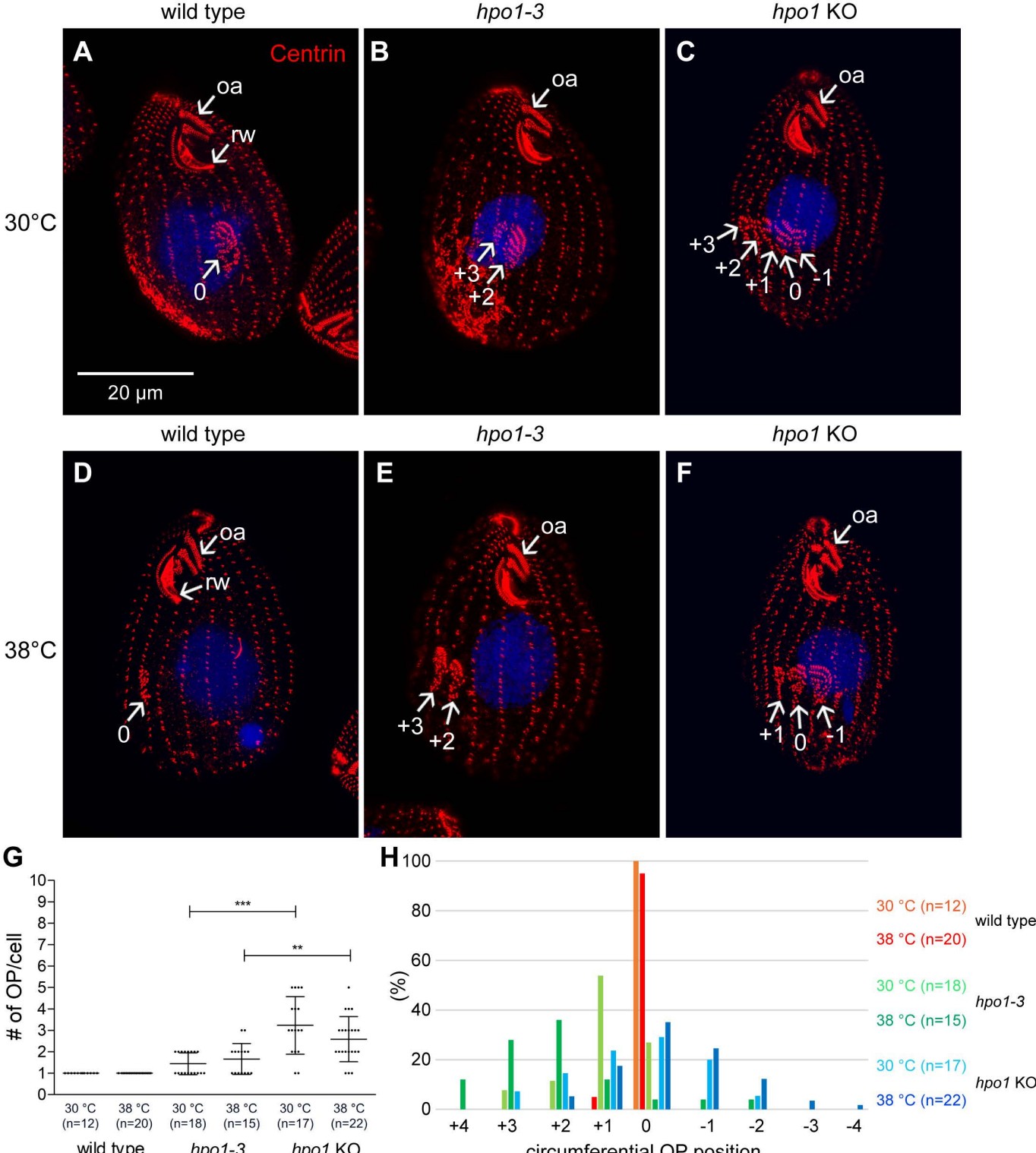

**Fig 2. *hpo1* alleles disturb the patterning on the circumferential axis by conferring an excessive number of oral primordia and their lateral displacement.** (A-F) SR-SIM images of cells labeled with the 20H5 anti-centrin antibody (red) and DAPI (blue). The cells were cultured overnight at either 30°C (A-C) or 38°C (D-F). (G,H) The graphs document an increase in the number of oral primordia (G) and their lateral displacement (H) conferred by

PLOS Genetics | https://doi.org/10.1371/journal.pgen.1011735   June 2, 2025

5 / 26

the *hpo1* alleles. Mean +/- SD. Two samples of cells were used for each genotype and the data were combined. The total number of analyzed cells is indicated (n=). A two-tailed unpaired t-test was executed for statistical significance (**: P<0.01; ***: P<0.001; ns, not significant). Abbreviations: oa, oral apparatus; rw, ribbed wall of microtubules. The arrows with numbers mark the C positions of the ventral rows with adjacent OPs.

*hpo1* mutants, the OP positions are not random. Rather, the *hpo1* phenotype can be described as an inability to focus to OP position to a single ventral row. Also as described [16], the severity of the *hpo1* allele phenotypes increases at 38°C as compared to the standard temperature 30°C (Fig 2G, 2H). As oral development progresses, adjacent OPs frequently fuse into one oral field and consequently most mutant cells have a single, mature OA (labeled oa in Fig 2B, 2E). Intriguingly, the *hpo1* alleles also reduce the number of CVPs per cell: while most wild-type cells have two CVPs (near two adjacent rows on the right side, usually +4 and +5), *hpo1* mutant cells often have only one CVP [16], (S1B Fig compare to S1A and S1H Fig).

We used comparative NGS to map genomic variants that co-segregate with each of the four available *hpo1* alleles (*hpo1-1*, *hpo1-2*, *hpo1-3* and *hpo1-4*). Linkage peaks were consistently detected on the micronuclear chr3 around ~5Mb (see the data for *hpo1-3* in Fig 3A), in agreement with the previous assignment of *hpo1-1* to chr3 using complementation tests in crosses between mutant homozygotes and micronuclear nullisomic strains [16]. Within the linkage region, we located two homozygous variants: chr3:5363773 G/A in *hpo1-1* and *hpo1-4* and chr3:5364108 T/C in *hpo1-2* and *hpo1-3* genomes, respectively. Both variants are located within the same protein-coding gene, *TTHERM_001276421*. BlastP searches identified TTHERM_001276421 homologs in diverse ciliate species including another oligohymenophoran, *Paramecium tetraurelia* (GSPATG00001303001 and 6 paralogs), and more evolutionarily-distant ciliates: the hypotrich *Oxytricha fallax* (g7224, g191167 and g3871) and the heterotrich *Stentor coeruleus* (SteCoe_23425, SteCoe_24205). While BlastP searches failed to identify proteins with significant homology outside of the ciliate phylum, the domain organization of TTHERM_001276421 resembles that of the conserved ciliary tip protein ARMC9, whose mutations cause Joubert syndrome [17–19]. Both ARMC9 and TTHERM_001276421 have the same set of protein domains arranged in the same order (Fig 3B-B'). The N-terminal one-third of TTHERM_001276421 is classified by Interpro as the ARMC9 domain (IPRO040369), within which there is a LisH domain (IPR006594), and two short coiled-coil regions (Fig 3B-B'). The C-terminal region of TTHERM_001276421 contains an ARM-like domain (IPRO11989) (Fig 3B-B'). Thus, it appears that TTHERM_001276421 is a ciliate lineage-specific ARMC9-like protein. The chr3:5363773 G/A variant (*hpo1-1*, *hpo1-4*) results in the S236N amino acid substitution, while the chr3:5364108 T/C variant (*hpo1-2*, *hpo1-3*) results in the F318S substitution, both within the ARM-like domain. In agreement with our findings, homozygotes for *hpo1-2* and *hpo1-3* were reported to have a similar temperature-sensitive phenotype, less severe than that of the *hpo1-1* and *hpo1-4* homozygotes ([16], and information deposited at the *Tetrahymena* Stock Center (https://tetrahymena.vet.cornell.edu/display.php?stockid=SD01466).

To test whether *TTHERM_001276421* is the locus of *hpo1* alleles, we used homologous DNA recombination to edit *TTHERM_001276421* (in the wild-type background) to encode the *hpo1-2(3)* linked F318S variant with a C-terminal 3xHA tag. Homozygotes expressing TTHERM_001276421-F318S-3xHA had the cortical phenotype similar to the one observed in the original *hpo1-3* mutants: multiple early-stage OPs forming near adjacent ventral rows (Fig 3D-D' compare to Fig 3C-C') or a single compound primordium (resulting from fusion of adjacent primordia) at a later cell division stage (Fig 3E-E'). While the wild-type TTHERM_001276421-3xHA protein was enriched near the anterior BBs along a subset of ciliary rows on the cell's right side (Fig 3C-C') (see below for a detailed analysis of the localization pattern), the F318S Hpo1 variant was either not-detectable above the background (Fig 3D-D') or greatly diminished but still biased to the cell's right side (Fig 3E-E' compare to Fig 3C-C'). Thus, F318S may reduce the targeting of TTHERM_001276421 to the cell cortex or decrease its stability. Next, we created a germ-line based strain with a deletion of *TTHERM_001276421*. The *TTHERM_001276421*-KO macronuclear homozygotes showed a phenotype similar to the original *hpo1* mutants with a

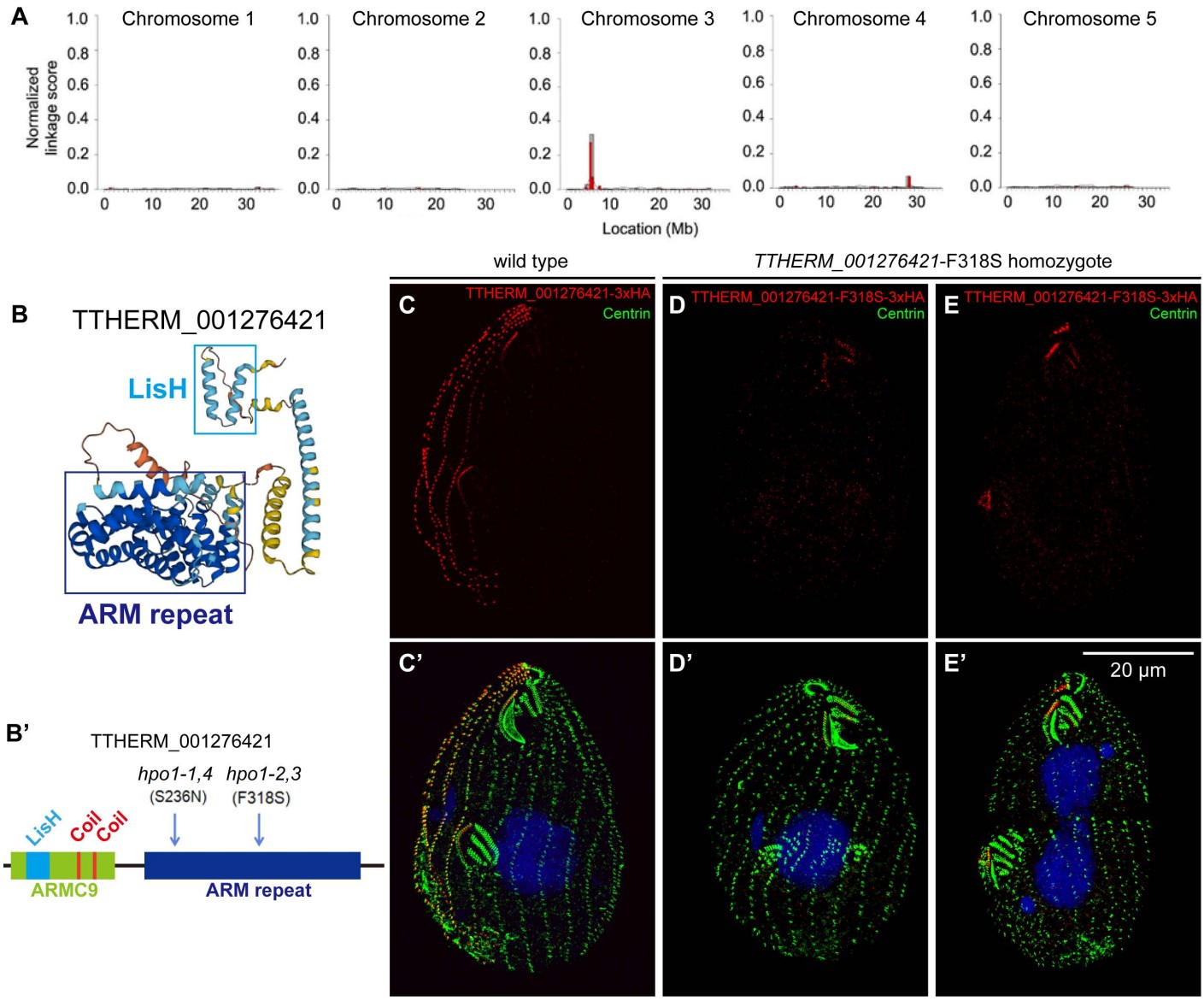

**Fig 3. *hpo1* alleles map to *TTHERM_001276421* encoding an ARMC9-like protein.** (A) Mapping of *hpo1-3* by the ACCA method. A linkage peak is present around 5 mB on the micronuclear chromosome 3. (B) The 3D structure of TTHERM_001276421 protein generated by AlphaFold2. (B') The domain organization of TTHERM_001276421 protein based on InterPro. The positions of two substitutions found in strains homozygous for the four *hpo1* alleles are marked. (C-E') SR-SIM images show the cortical localization of either TTHERM_001276421-3xHA (a C-terminally tagged wild type protein) (C,C') or TTHERM_001276421-3xHA with the *hpo1-3*-linked substitution, F318S (D-E'). Note that expression of the F318S variant of TTHERM_001276421 phenocopies the cortical organization (multiple and shifted OPs) of *hpo1-3*. After growth at 38°C for 3 hours, the cells were labeled with the anti-HA antibody (red), 20H5 anti-centrin antibody (green), and DAPI (blue).

subtle difference: the *knockout* cells had a higher number of OPs per cell as compared to the *hpo1-3* homozygotes (Fig 2C, 2F, 2G). Furthermore, while in the *hpo1-3* homozygotes, the OPs were predominantly shifted to the cell's right side, in the *TTHERM_001276421*-KO cells the OPs were more frequently shifted to the cell's left side at both 30 and 38°C (Fig 2C, 2F compare to Fig 2B, 2E, Fig 2H). A leftward shift was reported to occur in the *hpo1-2* homozygotes grown at 39°C for 24 hr [16]. Thus, the direction of the shift in the stomatogenic row position appears to depend on the degree of loss of

function of Hpo1 and the original *hpo1* alleles are likely hypomorphs. Overall the phenotypes conferred by the engineered alleles are remarkably similar to the phenotypes observed in the original *hpo1* mutants [16] and therefore we concluded that *TTHERM_001276421* is the sought *HPO1* gene.

## Hpo1 forms a circumferential cortical gradient with a high point on the cell's right side

During interphase, Hpo1-3xHA was enriched near the most anterior BBs of ~6 somatic ciliary rows on the cell's right side (Fig 4A-A'''). Along these rows, the levels of Hpo1-3xHA were high within the first most anterior 8-10 BBs and decreased along row length (Fig 4A-A'''). In addition, Hpo1 marked the BBs of the UM row positioned on the right side of the OA (labeled um in Fig 4A-A'). On the ventral side, there was an abrupt drop in the Hpo1 level between row + 1 and row 0 (Fig 4A-A'). This seems important because the OP starts to form in the immediate proximity of BBs of row 0 (likely by nucleation from somatic BBs of row 0 serving as templates [20]). In the early divider with a young OP, the Hpo1-3xHA pattern was unchanged, with a drop-off between rows +1 and 0 and the presence in UM (Fig 4B-B'''). In dividers with a fission line, the pattern of Hpo1-3xHA was duplicated along the AP axis (Fig 4C-C'''). Namely, Hpo1-3xHA appeared along the anterior BBs of a subset of posterior ciliary half-rows on the right side of the cell, mirroring the positions of the pre-existing Hpo1-3xHA in the anterior hemi-cell (Fig 4C-C'''). Thus, the C pattern of Hpo1 is faithfully replicated during the cell cycle. Observations of live cells expressing GFP-Hpo1 revealed a pattern of Hpo1 distribution consistent with the data obtained in fixed cells (S2B, S2C Fig). In addition to the localization at the BBs, Hpo1 weakly localized at positions consistent with the microtubule bundles associated with the BBs (transverse and longitudinal microtubule bundles, S2A–S2C Fig). We did not detect displacements of GFP-Hpo1 foci in live cells.

A close inspection of the levels of Hpo1-3xHA around the cell's circumference revealed a second drop-off on the cell's dorsal side (Fig 4A''-A''', 4B''-B'''). In the cells stained for fenestrin, a marker of CVPs [21,22], the dorsal drop-off was apparent between the second and third row past the CVP row counting clockwise (if the cell was viewed from the apical end) (Fig 5C'). To map the position of the dorsal Hpo1 drop-off more accurately, we imaged the cell's apical region in cell fragments that offered a "polar" view of the circumference (Fig 4D, 4E). The centrin signal was nearly uniform around the cell circumference (Fig 4D, 4E). On the dorsal surface, the Hpo1-3xHA signal was highest at row + 4 and decreased to almost the background level in two steps: between row + 5/ + 6 and +7/ + 8. (Fig 4D, 4E).

Overexpression of Hpo1-3xHA using the cadmium-dependent promoter *MTT1* [23], resulted in strong accumulation in the cell body. However, the right-side- and anterior-biased localization of Hpo1 was still visible and the cortical pattern was not disturbed (S3 Fig). Likely, the biased localization of Hpo1 requires its binding to another spatially-biased cortical factor, and Hpo1, while required, is not rate-limiting for precise positioning of oral development.

## The dorsal Hpo1 drop-off marks the position where an extra oral apparatus assembles in the *janus* C (*janC-4*) mutant

In the *janus* mutants (*janA*, *janB* and *janC* allelic groups), ventral structures are abnormally duplicated on the dorsal cell surface [24–27]. The dorsal (secondary) OA is underdeveloped and often inverted in its left-right internal polarity (sOA in Figs 5B and S4A). The *janus* phenotype was interpreted as a global mirror-image circumferential pattern duplication [24,25,27]. In the *janC-4* homozygotes, the distribution of Hpo1-3xHA was similar to that in the wild type, including an enrichment of Hpo1 on the cell's right side (Fig 5A-B' compare to Fig 4). The average number of rows with high Hpo1-3xHA (rows between the two drop-off positions) was elevated from 7 in the wild type to 8 in the *janC-4* homozygotes (Fig 5E), but *janC-4* cells had more total rows per cell and therefore the ratio of Hpo1-enriched rows to the total cell circumference was unchanged (Fig 5F). The expression of the sOA in *janus* mutants is partially-penetrant [25–27]. Both *janC-4* mutant cells with and without a sOA had a similar number of Hpo1-enriched rows (Fig 5B-B' compare to Fig 5A-A'). Strikingly, in cells with a fully expressed *Janus* phenotype, the sOA was located at the position of the dorsal

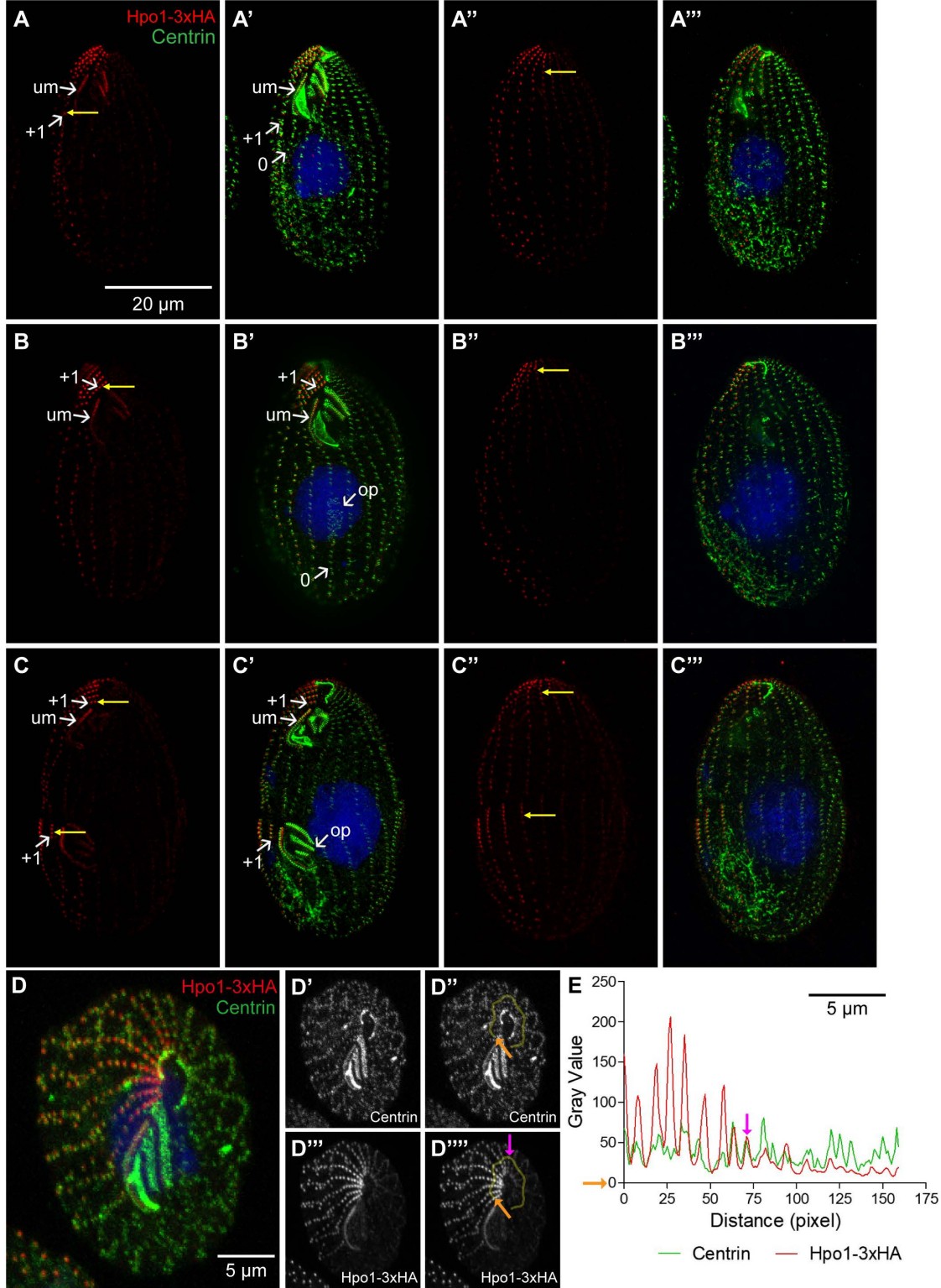

**Fig 4. Hpo1 forms a bidirectional gradient with a high point on the cell's right lateral side.** (A-C''') SR-SIM images showing two sides of cells expressing Hpo1-3xHA during the cell cycle. Cells were labeled with the anti-HA (red) and 20H5 anti-centrin (green) antibodies, and DAPI (blue). (D-D''') Confocal image of an apical fragment of a cell expressing Hpo1-3xHA. The cell was labeled with anti-HA (red) and 20H5 anti-centrin (green) antibodies,

and DAPI (blue). Single channel gray scale images for centrin (D',D") and Hpo1-3xHA (D''',D''") are shown. **(E)** The intensity plots document the gray values measured across the region marked by the yellow lines shown in panels D" and D"." The orange arrows indicate where the measurements were started, and the purple arrow indicates the position of the dorsal discontinuity. Abbreviations: um, undulating membrane row in OA; op, oral primordium. The numbers mark the positions of ventral rows. Yellow arrows mark the Hpo1 "contrast row", a row with high Hpo1 level next to a row with a drop off in Hpo1 level.

discontinuity of Hpo1 (next to row + 7 in Fig 5B-B'; S4 Fig). Thus, in the *janus* background, Hpo1 is a bilateral marker for oral development.

The *janus* mutants also have an increased number of CVPs that often are arranged as two sets separated by 1-2 rows [24,25]. Thus, there is a correlation between the widening of the Hpo1-enriched domain and the increased number of CVPs in the *janC-4* homozygotes. The mid-point of the CVP domain was located along or close to the row with the peak level of Hpo1 in both the wild type (Fig 5C') and *janC-4* cells (Fig 5D').

To summarize, OA development occurs at positions where Hpo1 levels drop off, while CVPs form at the peak level of the Hpo1 circumferential gradient.

### Hpo1 acts bidirectionally to restrict circumferential positions competent for oral development

Based on its circumferential distribution pattern in the *janC-4* homozygotes, Hpo1 may act bidirectionally to exclude OPs from the region of its enrichment on the cell's right lateral side. To test this idea, we analyzed double mutant homozygotes, *janC-1;hpo1-3.* The double mutants presented a highly penetrant phenotype: two mature OAs were located side by side near the anterior cell end, sometimes separated by a gap of one or more ciliary rows (Fig 6D compare to Fig 6A–6C). While such cells were rare in the single *janC-1* or *hpo1-3* mutants, the frequency reached 70% in the double mutants (Fig 6I, 6J). Given its high frequency, the "twin OA" configuration appears relatively stable and is likely propagated during division. Some twin OAs appeared to integrate into a single, compound organelle (Fig 6E). The twin OAs likely result from the "cortical slippage": lateral shifts in the OP positions in relation to the positions of the parental OAs. Likely both the primary and secondary OAs undergo cortical slippage and eventually posterior cells are produced in which two OAs collide (a situation predicted in [14]) (Fig 6K). The high frequency of twin OAs suggests that the circumferential direction of slippage is opposite for the primary and secondary OAs. We observed dividing *janC-1;hpo1-3* cells in the course of "cortical slippage" where the distance between the OAs decreased within one generation; the dividing cells in Fig 6F–6H have two old OAs still separated by a gap, and multiple immediately adjacent OPs. When the parental OAs are relatively close, they are mostly correct in structure suggesting the secondary OP almost invariably underdeveloped (an often inverted) in the single janus mutants, normalizes when it moves toward the primary OA in the double mutants. It appears that the predominant direction of cortical slippage is opposite: toward the cell right for the primary OP, and to the cell left of the secondary OP as depicted in Fig 6K. However, we can not exclude the possibility that the two OAs undergo slippage in the opposite directions (primary OP to the left and secondary OA to the right) or that the slippage directions are variable. Overall, these data suggest that Hpo1 acts to separate the two regions competent for oral development, one of which (dorsal) is repressed in the wild type through action of the *Janus* gene products. Furthermore, these data correlate the bilateral gradient pattern of Hpo1 (described above) with its bilateral (OP excluding) activity.

### Hpo1 interacts genetically with Bcd1, a left side-enriched OP positioning factor

Bcd1 is a conserved Beige-BEACH domain protein [8] whose loss of function in *Tetrahymena* confers a phenotype similar to that of *hpo1*: multiple adjacent oral primordia [15]. However, while the *hpo1* alleles reduce (S1A–S1C, S1H Fig), *bcd1* alleles increase the number of CVPs per cell, respectively [8,15] and (S1F, S1H Fig). Importantly, while Hpo1 is enriched

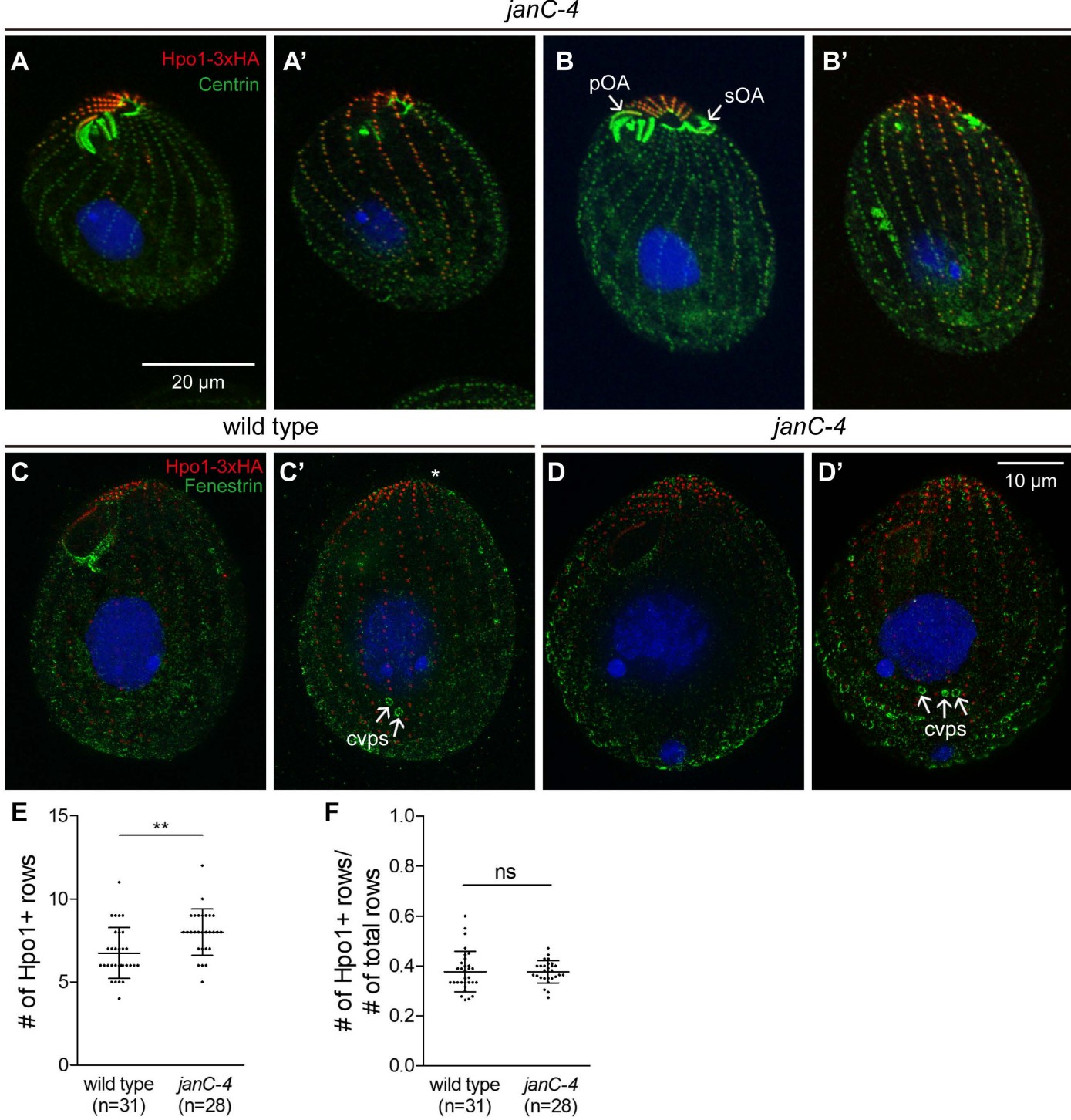

**Fig 5. The dorsal discontinuity in the Hpo1 gradient marks the cryptic position for oral development expressed in the *Janus* C mutant.** (A-B') Pairs of confocal images showing two sides of *janC-4 homozygote* cells that have either a single OA (A'A') or two OAs (B-B') (note that the *janC-4* phenotype is not fully penetrant). Cells were labeled with the anti-HA (red) and 20H5 anti-centrin (green) antibodies, and DAPI (blue). (C-D') SR-SIM image pairs of cells that are either wild-type (C, C') or *janC-4* homozygous (D, D') and express Hpo1-3xHA. The cells were decorated with the anti-HA (red), anti-fenestrin (green) antibodies, and DAPI (blue). (E, F) Graphs reveal an increase in the number of rows with enriched Hpo1-3xHA in the *janC-4* background as compared to the wild type (E) but the fraction of the circumference occupied by high Hpo1-3xHA remains unchanged when the increase in the

total number of rows is taken into account **(F)**. Mean +/- SD. Two samples of cells were analyzed for each genotype and the data were combined. The number of analyzed cells is indicated (n=). A two-tailed unpaired t-test was executed for statistical significance (**: P < 0.01; ns: not significant). Abbreviations: pOA, primary oral apparatus; sOA, secondary (dorsal and usually partially assembled) OA; cvps, contractile vacuole pores.

on the cells' right side, Bcd1 is enriched on the cell's left side, where it appears to also form a bidirectional circumferential gradient when viewed toward the apical cell end (See Fig 5H in [8], and S5A–B''' Fig). The AP distribution of Bcd1 protein resembles that of Hpo1, high near the most anterior row ends and fading away toward the posterior row ends ([8] and S5A–B''' Fig).

The similarity of the *hpo1* and *bcd1* OP phenotypes and the enrichment of Hpo1 and Bcd1 proteins on the opposite sides of the stomatogenic ciliary row suggest that the two proteins act collectively to restrict the OP formation from spreading to either the cell's right (Hpo1) or cell's left (Bcd1) of row 0. To look for interactions between Hpo1 and Bcd1, we first examined whether a deficiency of Bcd1 (*bcd1-2*) affects the pattern of Hpo1-3xHA (Figs 7, 8). In the *bcd1-2* homozygotes, the average number of Hpo1-enriched rows (rows between the discontinuities) was increased to 9 (Fig 8F) but the total number of rows per cell also increased and therefore Hpo1-enriched domain was proportionally unchanged (Fig 8G). As in the wild type, in the *bcd1-2* homozygotes, Hpo1-3xHA showed a drop-off in concentration both ventrally and dorsally (Fig 7B-B' compare to Fig 7A-A'; top view in Fig 7D, 7E compare to Fig 4D, 4E). However, the *bcd1-2* background altered the AP distribution of Hpo1-3xHA. In the wild type, the anterior-posterior Hpo1-3xHA gradient is steep, dropping to half signal intensity within several BBs of the anterior cell end. In *bcd1-2* homozygotes, the AP distribution of Hpo1-3xHA was more uniformed, only dropping to ~75% signal intensity over the entire row length (Fig 7A", B", C).

Next, we examined how the position of the ventral discontinuity of Hpo1 (that marks the position of normal oral development) is affected by *bcd1-2*. In wild-type cells, the ventral Hpo1 drop-off occurs between row + 1 (contrast row) and row 0 (Fig 8A-B'). In the dividing *bcd1-2* cells in which the OP fields were shifted laterally, the contrast row position was consistently shifted to the row on the right side of the right-most OP. For example, in the dividing cell shown in Fig 8D-D', there are two early OPs at positions 0 and +1 and the contrast row is located to the right of the OP pair at row + 2. While we have not observed clear cases of a shifted contrast row in interphase cells, there were instances of apparent ambiguity in position of the contrast row. In the *bcd1-2* mutant cell shown in Fig 8C-C', within the anterior 1/3 of the cell the contrast row is + 1. However, in the posterior 2/3 of the cell the contrast row is + 2 (yellow arrows). This cell has 3 postoral rows, a feature common to both *bcd1* and *hpo1* mutants that correlates with the increased width of the mature OA. However, another interphase *bcd1-2* cell shows the same "ambiguous contrast row" phenotype in the presence of two postoral rows (yellow arrows in Fig 8E-E'). Taken together, these data reveal that the Bcd1 deficiency modifies both the AP and C distribution of Hpo1. These observations suggest that there is a cross-talk between Hpo1 and Bcd1, possibly as a boundary interaction between the right and left ventral cortical region. No obvious effect of expression of *hpo1-3* on the distribution of Bcd1-GFP was observed but the signal of Bcd1-GFP was weak even in the wild-type background making the evaluation difficult (S5 Fig).

We next asked whether there is a genetic interaction between Hpo1 and Bcd1 by analyzing the phenotypes of double mutants homozygous for *bcd1-2* and *hpo1-KO* alleles. Surprisingly, the *bcd1-2;hpo1-KO* homozygotes were not viable. To determine the penetrance of synthetic lethality, we attempted to rescue the progeny of mating (double mutant) heterokaryons by biolistic introduction of a transgene expressing Hpo1-3xHA in an unrelated locus. A total of about 5 x 10⁶ mating double mutant heterokaryon cells were either subjected to biolistic bombardment with a Hpo1-3xHA transgene or mock-transformed without plasmid DNA and selected with paromomycin (conferred by the *neo* gene embedded in the *hpo1-KO* allele). Eighteen drug-resistant clones were isolated from the population bombarded with the Hpo1-3xHA transgene and one clone was isolated from the mock-transformed population. Six clones from the plasmid bombarded

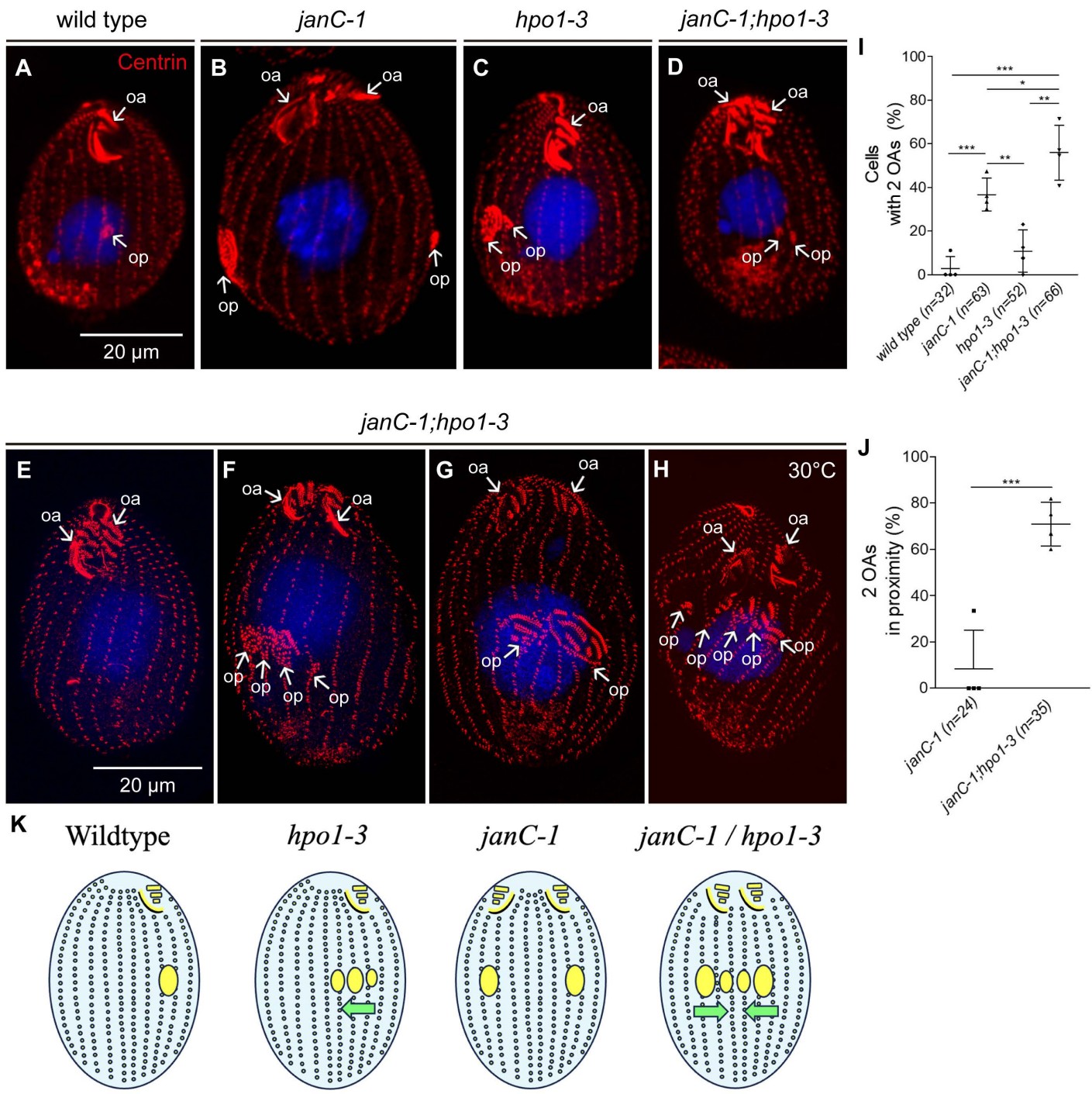

**Fig 6. Hpo1 acts bidirectionally to separate the two oral apparatuses in the *janus* C mutant.** Confocal (A-D) and SR-SIM (E-H) images cells with indicated genotypes labeled with the 20H5 anti-centrin antibody (red), and DAPI (blue). Cells shown in all panels except H were incubated for 4 hours at 38°C to enhance the *hpo1-3* phenotype. **(I-J)** The graphs quantifies the frequencies of cells with two mature OAs (I) and two OAs in close proximity **(J)**. Mean +/- SD. N = 4 experiments. Between 6-27 cells were analyzed per experiment and per genotype. A two-tailed unpaired t-test was executed for statistical significance. Stars indicate a statistically significant difference (*: P<0.05, **: P<0.01, ***: P<0.001, ns: not significant). The total number of cells per genotype is indicated (n=). **(K)** The drawing illustrates the likely outcome of the genetic interaction between *janC-1* and *hpo1-3*. The yellow ovals are OPs. The green arrows shows the proposed predominant direction of shifts in the OP positions. Abbreviations: pOA primary OA; sOA secondary OA; op, oral primordium.

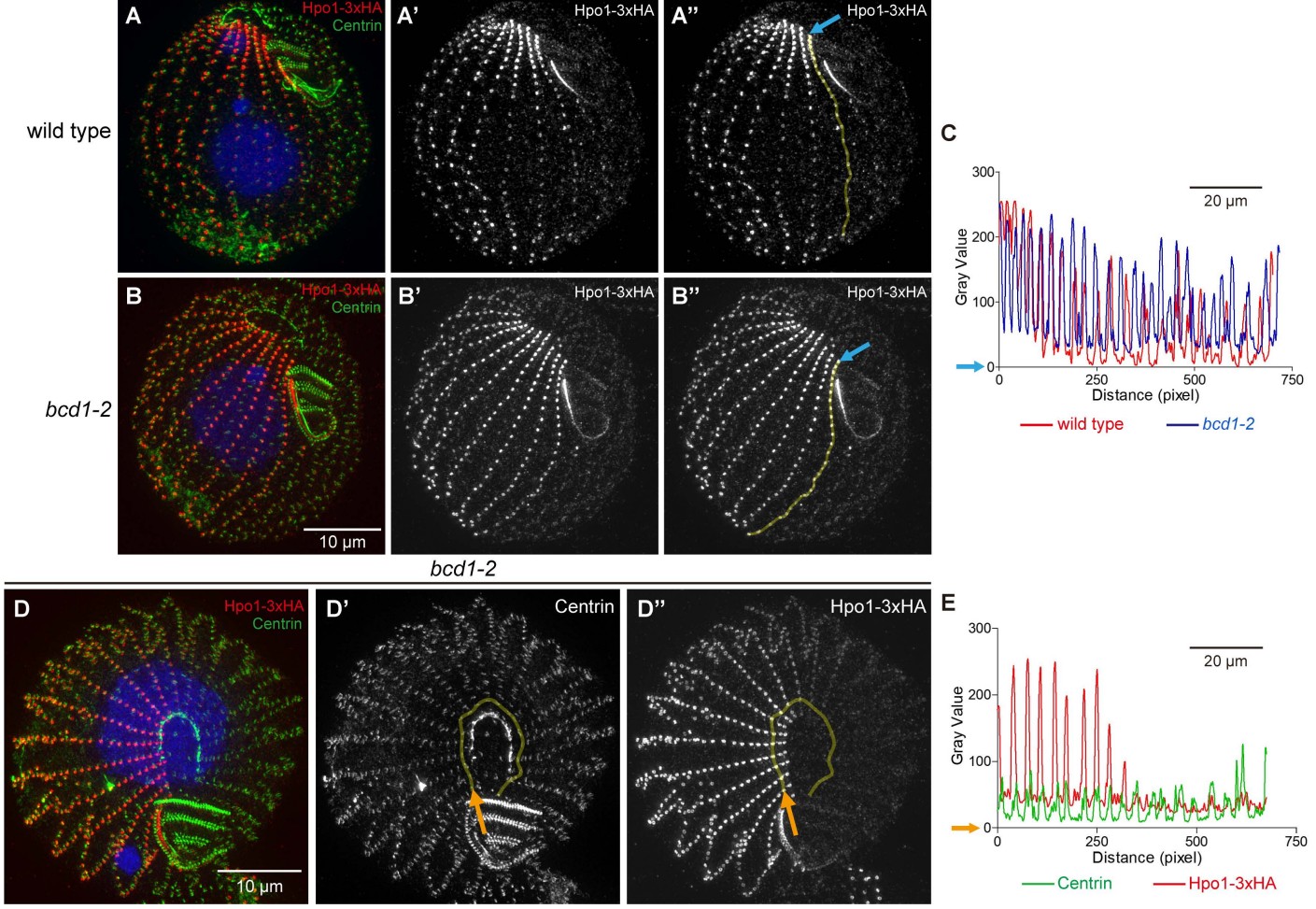

**Fig 7. A Bcd1 deficiency changes the A/P distribution of Hpo1.** (A-B") SR-SIM images of Hpo1-3xHA-expressing cells that are either otherwise wild-type (A-A") or *bcd1-2* homozygous (B-B"). The cells were labeled with the anti-HA (red) and 20H5 anti-centrin (green) antibodies, and DAPI (blue). The grey scale images are the single channel signals of Hpo1-3xHA. The yellow lines (A" and B") cover the region whose gray values were measured along the A/P axis, and the light blue arrows indicates the initial measurement. The corresponding A/P grey value intensity plots for Hpo1-3xHA in the two genetic backgrounds are shown in the graph in panel C. (D-D") An SR-SIM image of the apical region of a cell expressing Hpo1-3xHA (red) in the *bcd1-2* background, also stained with anti-centrin antibody (green). The single channel grey scale images for centrin (D') and Hpo1-3xHA (D") were used for measurements of the circumferential signal intensities. The measured regions are marked by the yellow lines and the orange arrows orient the measurements. The resulting signal intensity plots are shown in the graph in panel E. Compare to the similar images of the wild-type apical region in Fig 4D, 4E.

cells were analyzed by immunofluorescence and all were positive for Hpo1-HA (S6B-C' Fig). The single clone selected in the mock-transformed population was negative for Hpo1-HA as expected (S6A-A' Fig). A PCR amplification detected a wild-type Hpo1 sequence corresponding to the portion that was deleted in the *hpo1*-KO allele in the "escapee" clone (S6D-D' Fig). Likely, in this single clone there was a transfer of the wild-type *HPO1* gene from the parental macronucleus to the newly developing macronucleus that rescued the lethality. These observations confirm that the loss of function of Bcd1 and Hpo1 is lethal and 100% penetrant (n = 5 x 10$^6$). Next, we evaluated how the cortical pattern changes during the transition from the wild-type to double mutant phenotype in the mass-selected progeny of mating heterokaryons. To this end, heterokaryon strains with complementary mating types were allowed to mate, refed and selected with paromomycin to kill the non-mating (phenotypically wild-type) parental cells. After 24 hr of drug selection, the double mutant progeny

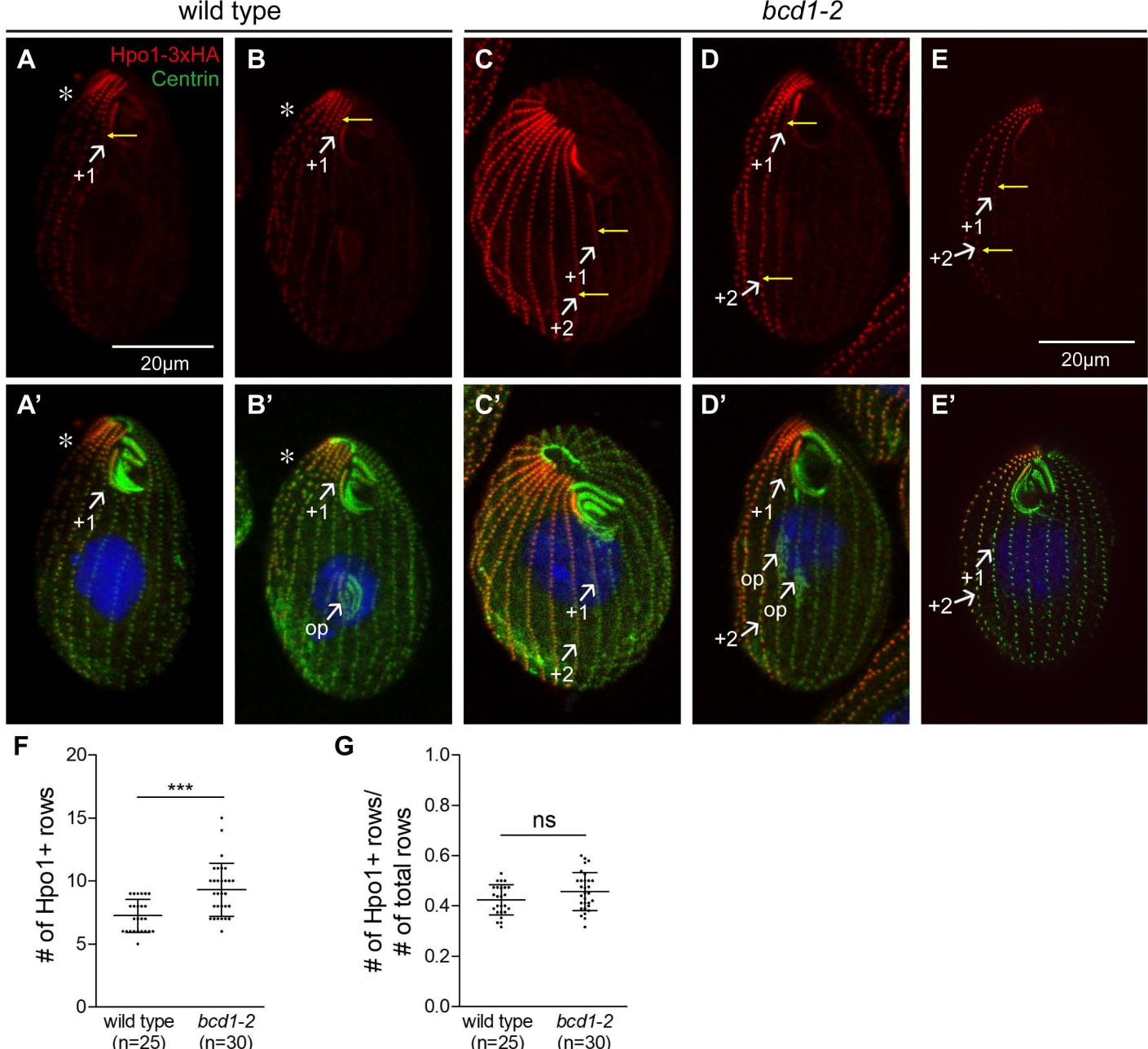

**Fig 8. Bcd1 influences the circumferential pattern of Hpo1-3xHA.** Confocal (A-D') and SR-SIM (E,E') images of cells expressing Hpo1-3xHA that are either wild-type (A-B') or homozygous for *bcd1-2* (C-E'). After overnight incubation at 30°C, the cells were labeled with the anti-HA (red), 20H5 anti-centrin (green) antibodies, and DAPI (blue). The numbers mark the ventral row positions. Yellow arrows mark the Hpo1 "contrast row", a row with high Hpo1 level next to a row with a drop off in Hpo1 level. Note the presence of two adjacent contrast rows on the ventral surface in the *bcd1-2* homozygotes (C-E, compare to A,B). **(F, G)** Graphs quantify the number of rows enriched in Hpo1-3xHA **(F)**, and the ratio of Hpo1 enriched rows to the total number of rows per cell **(G)**. Mean +/- SD. Cells were analyzed in 3 experiments and the data were combined. A two-tailed unpaired t-test was executed for statistical significance. Stars indicate a statistically significant difference (***: P<0.001, ns: not significant). The total number of cells per genotype is indicated (n=).

cells looked nearly normal except for the frequent OAs with abnormally wide oral M rows (white arrows in Fig 9A). After 30–46 hr, dividing cells frequently had posteriorly displaced primordia (yellow arrows in Fig 9B, 9E–9G). In the cells with a posteriorly shifted OP, the division boundary position was variable, either also shifted to the posterior (asterisks in Fig 9F) or equatorial (asterisks in Fig 9E) or shifted anteriorly (asterisks Fig 9G). Cells with a posteriorly shifted OP and an anteriorly shifted division plane may produce a posterior daughter cell with mature OA near the posterior cell end (Fig 9H). At ~46 hr the cells were uniformly arrested in cell proliferation and had an abnormally elongated and curved morphology with OAs that had curved (pink arrows in Fig 9C, 9I) or fragmented M rows (green arrows Fig 9C). The somatic ciliary rows appeared excessively long, and sometimes twisted (Fig 9D, 9H, 9I). SR-SIM imaging revealed additional patterning defects including OAs that lacked the UM (Fig 9G, 9I) and OAs with M rows arranged with orthogonal orientations (small arrows in Fig 9J, 9K). In addition, there were defects in the organization of the buccal cavity including a missing or fragmented ribbed wall of microtubules (Fig 9G, 9I–9K). The *bcd1-2;hpo1*-KO heterokaryon progeny were not viable on the specialized culture medium (MEPP) that supports proliferation of mutants lacking a functional oral apparatus [28]. Thus, the lethality is not solely caused by a lack phagocytotic activity that occurs exclusively at the buccal cavity of the oral apparatus. We conclude either Hpo1 or Bcd1 are required for survival and that both proteins contribute to patterning in the ways that extend beyond positioning of the OP on the cell-wide polarity axes, including shaping the internal organization of the OA and controlling the longitudinal expansion of the somatic ciliary rows.

## Discussion

Ciliates are well suited for studies on intracellular pattern formation but remain relatively unexplored. In *Tetrahymena*, mutations in several loci selectively affect organelle positioning along either the AP or C axes (reviewed in [2,14]), suggesting that the pathways operating on each of the two orthogonal polarity axes have a degree of independence. Several highly conserved kinases and kinase-binding proteins, including components of conserved Hippo signaling, mediate organelle positioning along the AP axis [29–34]. The emerging view is that on the AP axis, organelle locations are determined by proteins that mark cortical domains within which new structures are not permitted to form (reviewed in [2]). These inhibitory cortical factors are either permanent (Elo1/Lats [32]) or appear shortly before the new cortical structures form (CdaI/Mst [31], CdaA/cyclin E [29]). Here we made an advance toward understanding the far less explored mechanism of patterning around the cell circumference in *Tetrahymena*, by identifying the *HPO1* gene. The *hpo1* alleles disturb circumferential patterning by permitting oral development outside of the standard row 0. Over generations, the ongoing shifts in the positions of oral meridians (cortical slippage) may even produce mutant cells that have CVPs on the left cell's side, and therefore have an inverted overall "handedness" [16].

Hpo1 is a ciliate phylum-specific ARMC9-like protein. Mutations in the canonical ARMC9 cause Joubert syndrome, a neuro-developmental ciliopathy [18,19]. In *Tetrahymena* and mammalian cells, ARMC9 localizes to BBs and the tips of cilia [17,18,35]. Likely, Hpo1 evolved by neofunctionalization of the ancestral ARMC9, following a gene duplication that occurred before the emergence of diverse ciliate subclasses. ARMC9 acts as a scaffold to form a complex with additional conserved ciliary tip-enriched proteins, which plugs the plus end of microtubules and attenuates their assembly rate [36]. Besides the degree of homology, the only shared property of Hpo1 and ARMC9 is the localization to the BBs. We speculate that Hpo1 acts as a scaffold for proteins that inhibit assembly of BBs that are destined to form the OP, possibly by inhibiting the growth of plus ends of triplet microtubules. We recently identified another conserved ciliary protein, a Fused/Stk36 kinase CdaH, as a key regulator of positioning on the AP axis [34]. Ciliates belong to the clade of Alveolata that also includes dinoflagellates and apicomplexans. Among the alveolate protists, only ciliates assemble multi-ciliated arrays. Thus, in the course of emergence of ciliates from an ancestral biciliated alveolate [37–39], multiple ciliary proteins could have gained functions in the patterning of ciliary rows.

Hpo1 forms concentration gradients along both cell polarity axes. Remarkably, on the circumferential axis, Hpo1 forms a bilateral concentration gradient with the high point on the cell's lateral right side and sharp drop-offs along the longitudes

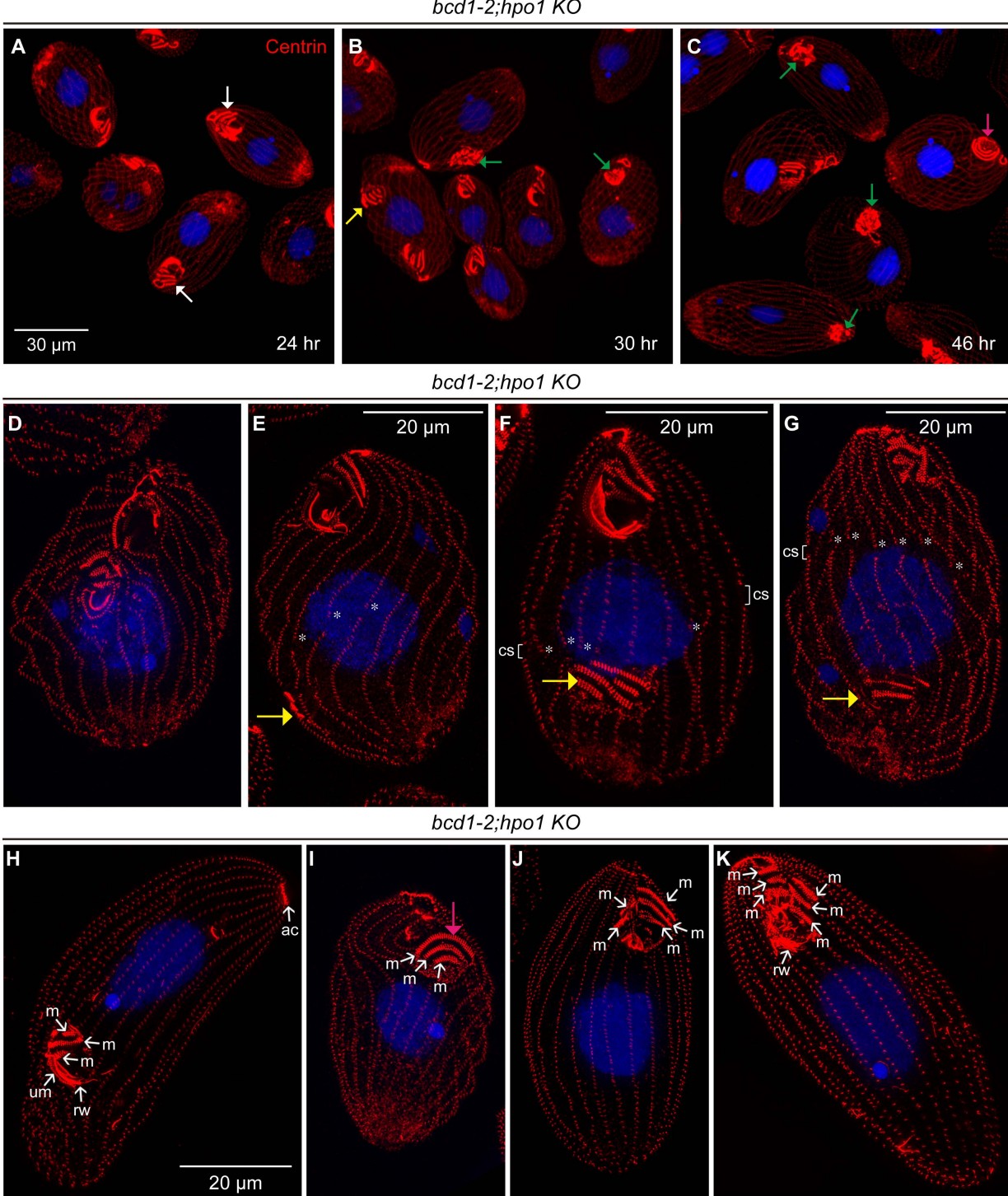

**Fig 9. A loss of both the left-side enriched Bcd1 and the right-side enriched Hpo1 severely disturbs cortical patterning.** Confocal (A-C) and SR-SIM (D-K) images of *bcd1-2;hpo1*-KO double homozygotes obtained as progeny of mating heterokaryons. The times shown in panels A-C refer to the period after refeeding of the mating heterokaryons (after 18 hr of mating in an inorganic medium at 30°C). Cells were labeled with the 20H5 anti-centrin antibody (red), and DAPI (blue). Abbreviations: m, M oral row; um, UM oral row; rw, ribbed wall of microtubules; ac, apical crown; cs, cortical

subdivision (fission line). The stars show the gaps in the longitudinal rows that indicate the position of the "cortical subdivision" (a set of gaps in rows at the division plane). Arrows of multiple colors indicate types of cortical defects: white, OAs with long oral rows; yellow, OP displaced posteriorly; green, OA with disorganized or fragmented rows; pink, OA with circular rows.

permissible of oral development. The pattern of Hpo1 appears invariable during the cell cycle. In this way, Hpo1 is reminiscent of Elo1, a Lats kinase that marks the posterior cell region and acts to prevent the OP from forming too close to the posterior cell end [32]. Thus, the positioning on both orthogonal polarity axes involves gradient-forming proteins that may function as pre-existing markers that define positions at which organelles assemble.

Important early insights into the mechanism of circumferential patterning were obtained using the giant ciliate Stentor coeruleus. *Stentor* species have formidable healing and regenerative capabilities. Fragments of *Stentor* lacking most of the cortex can regenerate a complete pattern and recover the ability to grow and divide, which argues that pre-existing cortical organelles are not an essential source of positional information for forming structures [40–42]. Importantly, in *Stentor*, the circumferential surface differentiation is apparent along the entire cell length. Unlike in *Tetrahymena*, in *Stentor*, the width of the inter-row spaces (called stripes) gradually increases around the cell circumference. The narrowest stripes are located on the right side of the ventral surface and the stripe width increases clockwise until the widest stripes meet the narrowest stripes within the ventral region named the "locus of stripe contrast" (LSC) [43,44]. OP develops at the LSC, within the narrow stripe zone and close to the margin of the wide stripe zone [44–46]. Remarkably, supernumerary OPs can be induced by cortical grafts that juxtapose a narrow stripe cortex with a wide stripe cortex, even if the boundaries of the graft would normally be far from the sites of cortical development [47]. Thus, oral development occurs at locations that are characterized by a "structural contrast". In *Tetrahymena*, Hpo1 forms a "molecular contrast" as the sharp level discontinuity between row +1 and the stomatogenic row 0. Topologically, the Hpo1-enriched region in *Tetrahymena* corresponds to the narrow stripe region in *Stentor*. Taken together, our observations and the grafting studies in *Stentor* suggest that the ventral "contrast region" contains both activating and inhibitory influences that control oral development. We speculate that the right lateral region, where Hpo1 is concentrated, is also a source of an unknown "oral activator" that is inhibited by Hpo1 (green gradient in Fig 1C). Hpo1 and the oral activator gradients may overlap for most of the length except at the positions where the Hpo1 levels drop, thus creating conditions permissible for oral development (Fig 1C). Consequently, (in cells having a normal Janus activity, see below) the OP can form at row 0.

By analogy to the emerging principle of positioning on the AP axis (where pattern regulators mark cortical domains where organelle assembly is inhibited), the primary activity of Hpo1 could be an exclusion of oral development from the cell's right lateral side. Indeed in all original *hpo1* mutants (expressing alleles that are likely hypomorphs), the extra primordia form more frequently to the right of row 0 ([16] and this study). Unexpectedly however, in cells with a knockout of *HPO1,* there was an increase in the frequency of extra primordia to the left of row 0. The *hpo1-2* mutants also occasionally form primordia on the left side and that the frequency of left-shifted OPs increases at 39°C ([16] and this study). We speculate that in addition to the exclusionary activity to the right of row zero, Hpo1 stimulates the exclusionary activity to the left of row 0, which requires Bcd1. Both Hpo1 and Bcd1 mutants assemble extra oral primordia on either side of row 0 [15,16]. It is tempting to speculate that Bcd1 and Hpo1 have a primary OP excluding activity in the areas of their enrichment (Hpo1 on the right and Bc1 on the left) and enhance each other's exclusionary activity as a boundary effect across row 0 (Fig 1C).

We show that Hpo1 is organized as a bilateral gradient with drop-offs on the dorsal and ventral side. Both drop-off positions mark sites for oral development in the *janC* homozygotes. *janA* and *janB* alleles confer the same *janus* phenotype as the *janC* alleles used here [48–50]. The JanA gene product was recently identified as a Polo kinase that localizes to the left-to-dorsal circumferential region, with a large overlap with Bcd1 and a partial overlap with Hpo1 on the dorsal

side [1] (see Fig 1C). Cole and colleagues propose that JanA represses the expression of ventral features on the dorsal side by interfering with a right-side enriched oral activator [1]. We show here that in the double mutant expressing *janC-1*, a loss of Hpo1 reduces the distance between the primary and the secondary OAs. This observation suggests that Hpo1 has bilateral activity, excluding oral development from both dorsal and ventral margins of its gradient. Hpo1 suppression of dorsal oral development is masked by the over-arching Janus suppression activity. The Janus gene products may either act to modify the parameters of the Hpo1 gradient (e.g., by reducing its steepness at the dorsal drop-off position) or suppress downstream components required for expression of oral structures. While in cells having wild-type Janus gene products, the right-sided influence of Hpo1 on the OP position is not expressed, on the dorsal side Hpo1 may engage in other positional activities, including its potential dorsal interaction with Bcd1 that may control the number of CVPs (see below).

Intriguingly, all gene products studied here, in addition to their effects on oral development, also affect the number of CVPs. While all alleles used here are either null or hypomorphs, *hpo1* alleles decrease while the *jan* and *bcd1* alleles increase the number of CVPs, respectively. During cell division, the OP forms before the CVPs, but it is unlikely that either the number or positions of OPs affect the number of CVPs, based on the phenotypes of *hpo1* and *bcd1* mutants that affect oral development in a similar way but have opposite effects on the number of CVPs. We observed that CVPs are located at the posterior ends of rows whose anterior ends are at or near the high point of the Hpo1 circumferential gradient. Furthermore, we show that a loss of either JanC or Bcd1 expands the Hpo1-enriched domain. Both Bcd1 [8] and JanA [1] are present in the dorsal region, potentially bordering or overlapping with Hpo1 and could act on the CVP rows indirectly by regulating the parameters of the Hpo1 gradient.

Unexpectedly, by combining null alleles *hpo1*-KO and *bcd1-2,* we uncovered that either the right-side-enriched Hpo1 or left-side-enriched Bcd1 are required for *Tetrahymena* survival. The double mutants lacking both Hpo1 and Bcd1 cease to proliferate and display diverse pattern defects. One of the prevailing phenotypes in the double mutants are frequent posterior shifts in the OP position. This phenotype was earlier seen in the *hpo1-2* mutants exposed to a higher temperature [16] and occasionally in the *bcd1-2* mutants as well (see Fig 4A in [15]). Frankel and colleagues suggested that Hpo1 may be a shared component of patterning pathways that operate on the C and AP axes [16]. We note however, that all C patterning factors studied to date (Bcd1, JanA and Hpo1) in addition to the circumferential distribution bias, also show a graded distribution along the AP axis ( [8,1], and this paper). Thus, there could be a deeper connection between the C and AP positioning pathways. The end point of the double mutant phenotype is an apparent cell cycle arrest with an enlarged mispositioned and disorganized oral apparatus and twisted somatic rows. The lethality of the double mutants is not caused by starvation, based on their inability to grow on the specialized medium (MEPP) that supports proliferation of mutants unable to phagocytose [51]. In ciliates, there is extensive cross-talk between the cell cortex and the nuclei. On one side, the cortical architecture influences the positions of nuclei in the endoplasm and the cell cycle progression [52–55]. On the other hand, *T. thermophila* cells that have lost their micronucleus, arrest in the cell cycle with an abnormally shaped cell cortex [56]. Also, a mutation in the micronuclear telomere repeat sequence causes a cell cycle arrest associated with excessive proliferation of BBs in the OP [57]. The end point phenotype of the *hpo1-KO;bcd1-2* double mutant cells resembles that of the *psm* (pseudomacrostome) conditional mutants, which at the restrictive temperature arrest with an highly enlarged OA and a twisted cell shape [58]. We speculate that in *Tetrahymena thermophila*, there exists a checkpoint mechanism that arrests the cell cycle when the oral apparatus is severely damaged.

To our knowledge, we are first to document essential interactions between the right and left side-enriched cortical pattern regulators. These observations remind about some fascinating cortical graft experiments performed on *Stentors* by Vance Tartar and Gotram Uhlig. When grafting created cortical boundaries with a relatively weak stripe contrast (e.g., obtained by juxtaposition of the left and right cortex after removal of most of either the dorsal or ventral surface), oral development was delayed and preceded by a period of remodeling along the heal lines that involved branching of a subset of wider stripes that generated narrowest stripes and consequently created a stronger "structural contrast" [44,47]

(reviewed in [13]). Thus, it appears that left and right cortex interact to generate cues that remodel the cell surface. Double mutants, lacking both Bcd1 and Hpo1, show defects not only in the positioning but also in the internal organization (including the left-right polarity) of oral structures. Our observations suggest that Bcd1 and Hpo1 are parts of the left-right cortical cross-talk that positions and shapes the ventral features including the oral apparatus.

To summarize, our observations suggest a multi-domain model for circumferential pattern formation in ciliates (Fig 1C). Multiple cortical regions are formed by bilateral gradients of patterning factors including Hpo1, Bcd1 and the Janus gene products. Adjacent or overlapping domains interact to regulate each other's distribution and to create boundary effects that generate positional cues for forming organelles.

## Materials and methods

### Strains and culture

All strains used were obtained from the *Tetrahymena* Stock Center (Cornell University, Ithaca NY, currently housed at Washington University, St Louis, MO USA; https://sites.wustl.edu/tetrahymena/). Cultures were grown in the SPPA medium (1% proteose-peptone, 0.2% dextrose, 0.1% yeast extract and 0.003% EDTA:ferric sodium salt) supplemented with antibiotics (SPPA medium) [59,60]. CU428 *mpr1-1/mpr1-1* (*MPR1; VII*)(TSC_SD00178) and B2086 *mat1-2/mat1-2* (*mat1-2; II*) (TSC_SD00709) were used as wild type controls. CU427 *chx1-1/chx1-1* (*CHX1; VI*) (TSC_SD00715) was used for outcrosses to map the *hpo1* mutations. B*VII (TSC_SD00023) was used for self-crosses. The following mutant strains were used: IA393 *hpo1-1/hpo1-1; eja1-1/eja1-1* (*hpo1-1, eja1-1; II*) (TSC_SD01463), IA418 *hpo1-2/hpo1-2* (*hpo1-2, VI*) (TSC_SD_01465), IA443 *hpo1-3/hpo1-3* (*hpo1-3; II*) (TSC_SD01455), IA480 *hpo1-4/hpo1-4* (*hpo1-4, II*) (TSC_SD01466), IA359 *janC-1/janC-1* (*janC-1, IV*) (TSC_SD00637), IA479 *janC-4/janC-4* (*janC-4; VII*) (TSC_SD01520), IA378 *bcd1-2/bcd1-2* (*bcd1-2*, V) (TSC_SD00641), IA437 *janC-1/janC-1* and *hpo1-2/hpo1-2* (*janC-1, hpo1-2(3); IV*) (TSC_SD01576). In addition, the following strains were made by editing the micronucleus by homologous DNA recombination (see below): UG20 *hpo1-F318S-3xHA-neo4/ hpo1-F318S-3xHA-neo4* (*hpo1-F318S-3xHA-neo4*), UG21 *hpo1::neo2/hpo1::neo2* (*hpo1::neo2*), and heterokaryon strains UG22 *bcd1-2/bcd1-2 hpo1::neo2/hpo1::neo2* (*BCD1, HPO1, pm-s*), UG23 *bcd1-2/bcd1-2 hpo1::neo2/hpo1::neo2* (*BCD1, HPO1, pm-s,* mates with UG22). To obtain progeny cells with the (terminally lethal) macronuclear genotype *hpo1::neo2, bcd1-2*, UG22 and UG23 heterokaryon strains were starved and allowed to mate for 24 hr at 30°C, the cell population was incubated in the SPPA medium for 6 hr followed by mass selection with paromomycin 100 μg/ml in SPPA to kill the parental cells.

### Mapping of *hpo1* alleles and protein structure prediction

We applied the ACCA method [31] to map the causal mutation for *hpo1-3* as described below (the same strategy was used to map the remaining three h*po1* alleles using appropriate homozygous strains). Strain IA443 was crossed to CU427 (homozygous for the cycloheximide (cy)-resistant allele *chx1-1* in micronucleus). The heterozygous F1 progeny (*hpo1-3/HPO1*; *chx1-1/CHX1*) was assorted in SPPA to cycloheximide (cy) sensitivity. A cy-sensitive F1 was mated to B*VII to produce F2 homozygotes using uniparental cytogamy [61]. F2 clones were selected with 15 μg/ml cy and their cortical phenotype was evaluated by immunofluorescence. Twenty-three phenotypically wild-type or mutant F2 clones were pooled, cultured in 25ml of SPPA overnight, and subjected to starvation in 60 mM Tris-HCl (pH 7.5) for 2 days at 30°C. Total genomic DNA was extracted from the starved F2 pools and used to construct genomic libraries using Illumina Truseq primer adapters. The libraries were sequenced on Illumina HiSeq X to obtain paired-end 150bp reads at 90x coverage. MiModD in the ACCA mode was used to identify variants linked to the *hpo1* phenotype as described in detail in [29,31]. A 3D model of Hpo1 was retrieved from the AlphaFold protein structure Database (https://alphafold.com/). Protein domains were identified by InterPro (https://www.ebi.ac.uk/interpro/).

## Gene editing in *T. thermophila*

To construct a plasmid for a genomic knockout of *TTHERM_001276421/HPO1*, fragments were amplified from the genomic DNA of the wild-type strain CU428 and subcloned on the sides of the *neo4* selectable marker using primer pairs: 5'-AATTCCGCGGCGAACTTC TGAGTCATCATTG-3', 5'-AATTCTGCAGCTTAAAGGCGTCTACCATTTTATTC-3' and 5'-ATTCCCGGGTCAAGTATTCAACTCCTCTAAGTG-3', 5'-AATTGGGCCCGAATACTCTGCTCGTGATGTCGA-3'. The resulting plasmid pcand2-KO-neo4 was design to target for deletion a portion of the coding region composed of 1383 bp starting at codon 250 and including 371 bp of the 3' UTR. Another targeting plasmid, pIA443-3HA was constructed to introduce the F318S substitution (*hpo1-3*) into the wild type background and simultaneously add a 3xHA epitope tag sequence at the 3' end of the coding region. To this end, *HPO1* fragments were amplified using the total genomic DNA from strain IA443 (homozygous for *hpo1-3*) using primers pairs: 5'-AATTCTCGAGGAACTGTATTAGGAGTGATTTCAC-3', 5'-TTAACTGCAGCATAAAATATCCAATTAATTTCAAATATCCACAAATATTATC-3' and 5'-AATTGGATCCTTAACTA ACTTCATCCTAGAAGCATTC-3', 5'-AAATACGCGTTTAGTTCAACACTTAGAGGAGTAGAATAC-3' and used to replace the gene targeting parts of the plasmid pIFT54-3HA-native-neo4 [62]. The targeting fragments of the above plasmids were released using restriction enzymes cleaving near the ends of flanking homologous sequences and the digested DNA was used for biolistic bombardment of mating CU428 and B2086 cells at an early stage of conjugation optimal for targeting genes in the micronucleus, followed by standard crosses to make mutant heterokaryons and homokaryons [63].

To prepare a strain for live imaging of Hpo1, the coding region of *TTHERM_001276421/HPO1* was cloned into the plasmid pGFP_PLK2-BTU1ov, downstream of the *MTT1* promoter and the GFP sequence. The primer pairs used to amplify *HPO1* were 5'-CTATACAAACGCGTGATGTAAAACTTACCTGATTGC-3', and 5'-GTTCGCTTACGGATCCTCATTAACTA ACTTCATCCTAGAAGC-3'. The transgene was placed between the UTR sequences of *BTU1* for targeting to this locus by homologous DNA recombination. The plasmid was digested by BamHI and MluI, biolistically introduced into the CU428 strain and transformants were selected with 100 µg/ml paromomycin.

To overexpress Hpo1-HA, a plasmid was made (pMTT1_Hpo1_HA) with the following sequence elements cloned between the UTR sequences of the *BTU1* gene: *Bsr* selectable marker, *MTT1* promoter, coding region of *HPO1* amplified with primers: 5'-TAAAATAATGGCCAAGTCGACAATGTAAAACTTACCTGATTGCG-3' and 5'-AACATCATAAGGATAAG CACCGGATCCTTAACTAACTTCATCCTAGAAGCATTC-3', HA epitope tag sequence. The targeting portion of the plasmid was released with SacI and BamHI, introduced biolistically into the *BTU1* locus (of CU428 strain) and transformants were selected with 60 µg/ml blasticidin S. To induce overproduction, the transgene-carrying cells were exposed to 2.5 µg/ml cadmium chloride for 6 hr.

## Microscopic imaging

*T. thermophila* cells were fixed and stained by immunofluorescence as described [29,60]. The primary antibodies used were: polyclonal anti-GFP (Rockland Immunochemicals, #600-401-215; 1:800 dilution), monoclonal anti-HA 16B12 (Covance; 1:300), polyclonal anti-HA (Proteintech, 51064-2-AP; 1:300) and monoclonal anti-centrin 20H5 (EMD Millipore; 1:200-300; [64]). The secondary antibodies were conjugated to either Cy3 or FITC (Jackson ImmunoResearch, 115-095-146 and 111-165-003; 1:300). The nuclei were stained with DAPI (Sigma-Aldrich). The labeled cells were mounted in 90% glycerol, 10% PBS supplemented with 100 mg/ml DABCO (Sigma-Aldrich). To image the apical surface (top view), cell fragments were obtained as follows: 1.5 ml of cell culture was concentrated at 2,800 rpm for 3 min, and washed with 1 ml of the nuclear isolation medium A (0.1M sucrose, 4% gum arabic, 0.0015M $MgCl_2$, 0.01% spermidine-HCl, pH 6.75) [65]. Cells were concentrated by centrifugation to 150 µl and combined with 160 µl of 1% paraformaldehyde/0.25% Triton X-100, followed immediately by addition of 1.9 µl of octyl alcohol. The mixture was vortexed for 10–60 seconds and 20 µl of the sample was air-dried at the room temperature on a cover glass. Next, immunofluorescence was conducted as described above. Images were collected on a Zeiss LSM 710 confocal microscope with a Plan-Apochromat 63×/1.40 oil DIC M27 objective and on an ELYRA S1 SR-SIM microscope equipped with a 63× NA 1.4 Oil Plan-Apochromat DIC. TIRF

microscopy was executed as described [66] except that partial immobilization of cells was achieved by entrapment in a small volume of culture medium.

## Statistical analysis

Using the GraphPad PRISM software, we executed two-tailed unpaired t-tests to evaluate differences. $P < 0.05$ was deemed statistically significant.

## Supporting information

**S1 Fig. Quantification of the number of CVPs per cell.** All genotypes are homozygous as indicated in the figure panels. (A-G) Confocal images of wild type (A), single mutants (B-D, F), and double mutants (E, G). The cells were labeled with the anti-fenestrin antibody (red), and DAPI (blue) after incubation for 4 hours at 38°C. The white arrows point to the CVPs. (H) The graph quantifies the number of CVPs per cell. The bars represent the mean ± SD. The number of cells scored is displayed in each of the bar. A two-tailed unpaired t-test was executed for statistical significance. ns: not significant. Stars indicate statistically significant (*: $P < 0.05$, **: $P < 0.01$, and ***: $P < 0.001$, ns: not significant). Red stars indicate statistically significant differences as compared to the wild-type. Abbreviations: pOA, primary OA; sOA, secondary OA (in the *janC-1* background).
(TIF)

**S2 Fig. *In vivo*, Hpo1 marks basal bodies and adjacent microtubule bundles.** (A) An illustration of *T. thermophila* with the magnified inset describing a portion of the cell cortex (reproduced with the publisher's permission from Fig 1 [67]. (B, C) Still images of live cells expressing GFP-Hpo1 that localizes at positions consistent with the basal bodies and microtubule bundles (indicated by arrows). Abbreviations: tm, transverse microtubule bundle; lm, longitudinal microtubule bundle; pc, postciliary microtubule bundle; bb, basal body; kd, kinetodesmal fiber (non-microtubular); m, membranelle M row; um, undulating membrane row; oa, oral apparatus; mic, micronucleus; mac, macronucleus; cvps, contractile vacuole pores.
(TIF)

**S3 Fig. Overexpression of Hpo1 does not affect the cortical organelle pattern.** Cells carrying a transgene expressing Hpo1-HA under the *MTT1* promoter were grown either without (A-D') or with addition of 2.5 μg/ml cadmium chloride for 6 hr (E-H'), fixed and labeled with the anti-HA (red) and either anti-centrin (A'-C'; E'-G') or anti-fenestrin (D', H') (green) antibodies and DAPI (blue). Note an accumulation of Hpo1-HA in the cell body of overproducing cells. Despite overproduction, the right-side and anterior gradients of Hpo1 are still apparent and the positions and number of OPs (E-G' compare to A-C') and the number of CVPs (H,H' compare to D,D') are unaffected.
(TIF)

**S4 Fig. Quantification of the circumferential distribution of Hpo1-3xHA in the *janC-4* homozygote.** (A-A'''') An SR-SIM image of Hpo1-3xHA and centrin in an apical cell fragment of the *janC-4* homozygote. (A) The cell fragment was labeled with anti-HA (red), 20H5 anti-centrin antibody (green), and DAPI (blue) after overnight incubation at 30°C. Duplicates of single channel grey scale images for centrin (A',A") and Hpo1-3xHA (A''', A'''') are shown. The yellow lines mark the area used for measurements of signal intensity. The pink arrows show the start locations for signal intensity measurements. The orange and cyan arrow indicates initial and end point of gap, respectively. (B) The graph shows that signal intensity plots for Hpo1-3xHA (red line) and centrin (green line).
(TIF)

**S5 Fig. Localization of Bcd1-GFP in *hpo1-3 homozygotes*.** (A-D''') Pairs of SR-SIM images showing two sides of the same cells that express Bcd1-GFP and are either otherwise wild-type (A-B''') or *hpo1-3* (C-D'''). The cells were labeled

with the anti-GFP antibodies (red), 20H5 anti-centrin antibody (green), and DAPI (blue) after a period of growth for 4 hours at 39°C.
(TIF)

**S6 Fig. The lethality caused by the loss of function of both Bcd1 and Hpo1 can be rescued by a transgene encoding Hpo1-3xHA.** (A-C') SR-SIM images clones selected from the population of mating heterokaryons homozygous in the micronucleus for *bcd1-2* and *hpo1*-KO alleles that were either subjected to a mock biolistic bombardment (without transgene DNA) (A,A') or biolistically transformed with a transgene encoding MTT1-Hpo1-3xHA (B-C'). The cells were labeled with anti-HA (red), 20H5 anti-centrin antibody (green), and DAPI (blue) after overnight incubation at 30°C. (D,D') The diagram shows the positions of PCR primers designed for amplification of the portion of *HPO1* gene sequence deleted in the *hpo1-KO* allele (D) and the gel image showing the PCR products amplified from the genomic DNA isolated either from the escapee clone or from rescued clone (D').
(TIF)

**S1 Data. Numerical data (Excel spreadsheet) that underlie the graphs in Figs 2G, 2H, 4E, 5E, 6I, 6J, 7C, 7E, 8F, 8G, S1 and S4B.**
(XLSX)

## Acknowledgments

J.G. gratefully acknowledges Joseph Frankel (University of Iowa) for providing the inspiration and encouragement to investigate the *hypoangular 1* mutant. The SR-SIM imaging was done at the Biomedical Microscopy Core, University of Georgia.

## Author contributions

**Conceptualization:** Jacek Gaertig.

**Formal analysis:** Wolfgang Maier.

**Funding acquisition:** Wolfgang Maier, Karl F. Lechtreck, Jacek Gaertig.

**Investigation:** Chinkyu Lee, Ewa Joachimiak, Wolfgang Maier, Yu-Yang Jiang, Karl F. Lechtreck, Jacek Gaertig.

**Methodology:** Wolfgang Maier, Karl F. Lechtreck, Jacek Gaertig.

**Project administration:** Jacek Gaertig.

**Supervision:** Jacek Gaertig.

**Validation:** Chinkyu Lee, Ewa Joachimiak.

**Visualization:** Chinkyu Lee, Mireya Parra, Eric S. Cole.

**Writing – original draft:** Jacek Gaertig.

**Writing – review & editing:** Chinkyu Lee, Ewa Joachimiak, Wolfgang Maier, Yu-Yang Jiang, Eric S. Cole, Jacek Gaertig.

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
