## [Decision Letter · Decision Letter 0]

4 Mar 2025

PGENETICS-D-25-00071

Left-right cortical interactions drive intracellular pattern formation  in the ciliate Tetrahymena

PLOS Genetics

Dear Dr. Gaertig,

Thank you for submitting your manuscript to PLOS Genetics. As you can see, the reviewers were positive about this work, but raised important points that you should consider. Therefore, we invite you to submit a revised version of the manuscript that addresses the points raised during the review process.

Please submit your revised manuscript within 30 days Apr 03 2025 11:59PM. If you will need more time than this to complete your revisions, please reply to this message or contact the journal office at plosgenetics@plos.org. Please include the following items when submitting your revised manuscript:

We look forward to receiving your revised manuscript.

Kind regards,

Gregory J. Pazour

Academic Editor

PLOS Genetics

Fengwei Yu

Section Editor

PLOS Genetics

Aimée Dudley

Editor-in-Chief

PLOS Genetics

Anne Goriely

Editor-in-Chief

PLOS Genetics

**Journal Requirements:**

2) Please note that the Author Summary should appear in your manuscript between the Abstract (if applicable) and the Introduction, and should be 150-200 words long. The aim should be to make your findings accessible to a wide audience that includes both scientists and non-scientists. Sample summaries can be found on our website under Submission Guidelines:

https://journals.plos.org/plosgenetics/s/submission-guidelines#loc-parts-of-a-submission

3) Your manuscript is missing the following section: Abstract.  Please ensure all required sections are present and in the correct order. Make sure section heading levels are clearly indicated in the manuscript text, and limit sub-sections to 3 heading levels. An outline of the required sections can be consulted in our submission guidelines here:

https://journals.plos.org/plosgenetics/s/submission-guidelines#loc-parts-of-a-submission 

5) We notice that your supplementary Figures are included in the main figures file. Please remove them and upload them with the file type 'Supporting Information'. Please ensure that each Supporting Information file has a legend listed in the manuscript after the references list.

Potential Copyright Issues:

i) Figure S2A. Please confirm whether you drew the images / clip-art within the figure panels by hand. If you did not draw the images, please provide (a) a link to the source of the images or icons and their license / terms of use; or (b) written permission from the copyright holder to publish the images or icons under our CC BY 4.0 license. Alternatively, you may replace the images with open source alternatives. See these open source resources you may use to replace images / clip-art:

7) Please note that your Data Availability Statement is currently missing the DOI/accession number of each dataset OR a direct link to access each dataset. If your manuscript is accepted for publication, you will be asked to provide these details on a very short timeline. We therefore suggest that you provide this information now, though we will not hold up the peer review process if you are unable.

8) Please amend your detailed Financial Disclosure statement. This is published with the article. It must therefore be completed in full sentences and contain the exact wording you wish to be published.

1) State what role the funders took in the study. If the funders had no role in your study, please state: "The funders had no role in study design, data collection and analysis, decision to publish, or preparation of the manuscript.

9)  Please ensure that the funders and grant numbers match between the Financial Disclosure field and the Funding Information tab in your submission form. Note that the funders must be provided in the same order in both places as well. Currently, the order of the grants is different in both places.

**Reviewers' comments:**

Reviewer's Responses to Questions

Reviewer #1: In the study entitled “Left-right cortical interactions drive intracellular pattern formation in the ciliate Tetrahymena”, Chinkyu Lee, Ewa Joachimiak, Wolfgang Maier, Yu-Yang Jiang, Karl F. Lechtreck, Eric S. Cole, and Jacek Gaertig identified, for the first time, the Tetrahymena gene HPO1 (TTHERM_001276421), which encodes a ARMC9-like protein responsible for the mutant phenotypes previously linked to Tetrahymena hypoangular 1 (hpo1) alleles. Homozygous hpo1 mutants exhibit multiple oral apparatus (OA) with abnormal positioning to the left or right of the normal site in the cell.

The authors provide strong evidence that hpo1 gene is responsible for these phenotypes by generating Tetrahymena strains carrying previously described hpo1 mutations and hpo1 knockout strains, both displaying similar phenotypes. They further localized the Hpo1 protein to ciliary basal bodies using a strain expressing Hpo1-GFP, or Hpo1-3xHA and assessed its overexpression phenotype. Their findings reveal that and Hpo1 protein preferentially localizes to anterior basal bodies along a subset of ciliary rows on the cell’s right side, with levels highest at the most anterior basal bodies and decreasing along the row length. These gradients are crucial for defining contractile vacuole pores positioning and ensuring OA exclusion from the right side of the cell.

Additionally, the authors explored genetic interactions between the genes bcd1 (cell’s left side-enriched factor) and janA (encodes a Polo kinase), which localizes to the left-to-dorsal circumferential region of Tetrahymena. They found that Hpo1 interacts with Bcd1, and the simultaneous loss of both proteins is lethal, severely disrupting OA positioning and organization. This interesting study builds on the authors' previous work on the gradient-forming factors involved in ciliate morphogenesis, as well as the maintenance and organization of complex cortical organelles during cell division.

The paper is worthy of publication; however, incorporating some minor suggestions will improve the manuscript.

• Figure 5 – In both confocal and SR-SIM images, it is challenging to observe the increased number of rows with enriched Hpo1-3xHA in the janC-4 background compared to the wild type, as shown in panel E. Therefore, these images may not be the most representative. For the confocal images, it would be helpful to include a wild type cell for comparison.

• Figure 7 – In panels (C-C’’), it would be helpful to include an SR-SIM image of a wild type cell’s apical region for comparison with the images of a cell expressing Hpo1-3xHA (red) in the bcd1-2background.

• A schematic similar to Figure 6K would be beneficial to summarize the findings from Figures 7 and 8, particularly regarding the genetic interaction between bcd1-2 and hpo1-3.

• It may be possible to combine Figures 7 and 8 if the confocal images from Figure 8 are moved to the supplementary figures. However, it is essential to clearly indicate in the panels that the cells in Figure 8 are homozygous for bcd1-2.

• Figure 9 – It would be helpful to indicate in the panels that the cells in the images are bcd1-2;hpo1-KO double homozygotes.

• Figure S4 – Although this information is provided in the legend, it would be helpful to indicate in the figure itself that the panels show janC-4 homozygotes.

• Figure S5 – As the authors mention, the Bcd1-GFP signal is indeed quite weak. However, it appears to be even weaker in hpo1-3 (panels C-D’’’). Is this observation accurate?

Reviewer #2: Summary:

In this manuscript entitled “Left-right cortical interactions drive intracellular pattern formation in the ciliate Tetrahymena” (PGENETICS-D-25-00071), the authors report the identification of the protein encoded by the gene mutated in hypoangular 1 mutants of Tetrahymena as a ARMC9-like protein. These mutants have defects in circumferential patterning detected by shortening of the distance between the oral apparatus and the contractile vacuole pores. This is caused by a progressively rightward shift in the formation of a new oral appartus each cell division.By using whole-genome sequencing and variant analysis of wild-type and mutant genetic segregants, the authors were able to convincing identify the HPO1 gene. The original mutant alleles appeared to be hypomorphs and a gene knockout caused a more penetrant/severe phenotype. The hpo1 protein was shown to localize to the cells right lateral side. The shift of the oral primordium in that direction suggests that normally, the hpo1 gene product acts negatively to restrict the formation of a new oral apparatus where hpo1 is present. The hpo mutants not only shift the position of oral apparatus, but exhibit multiple foci of oral basal body poliferation, similar to the bcd1 mutants. The authors show that the bcd1/hpo1 double mutant cells result in cell lethality. This synthetic lethal phenotype is significant as Bcd1 is localized to the region to the left of the oral primorium. The authors propose that the two gene products act synergistically to prevent new oral apparatus formation to the left and the right of the row topped by the old oral apparatus. This study provides one of the first identifications of gene that controls left-right asymmetry in Tetrahymena.

Strengths:

The study is carefully executed and the experiments well described. The identification of the hpo1 protein is one of the first descriptions of the action of a protein that controls circumferential patterning in these highly structured unicellular eukaryotes, Tetrahymena. The genetic interactions and detailed cellular localization of the hpo1 protein in both Janus A (JanA) and broaden cortical domains (Bcd1) proteins provides important insight into the interplay into these major patterning determinants. This study reveals that multiple proteins with specific localizations provides the groundwork to future studies to characterize the precise signaling events needed for a Tetrahymena cell to precisely produce a daughter cell each cell division.

Weaknesses/comments to address:

1.There is little discussion as to how this ARMC9 protein may act to repress oral formation. Given that ARMC9 proteins are found in basal bodies and cilia in mammalian cells, the authors postulate that its role in circumferential patterning may be neo-functionalization. As a genetic definition, this is sufficient, but we don’t really learn of any possible molecular action through their findings or discussion.

2. Page 12. The heading states that Hpo1 interacts with Bcd1. As the results reveal a synthetic lethality/genetic interaction, I would suggest making the heading reflect the data. When this reviewer reads X interacts with Y, I expect of see some protein-protein interaction. Please provide a more specific heading.

3. Page 19. In the discussion of the JanC/Hpo1 double mutant phenotype, i.e. the primary and secondary oral apparati are closer, the authors indicate that suggest that the hpo1 gene product likely restricts the position of oral appartus formation of both the primary and secondary oral apparati. From the images shown in figure 6, it was not clear whether both the position of primary and secondary oral apparati are shifted equally.

4. It would be interesting to discuss why the Hpo1 and Bcd1 show synthetic lethality but not the JanC and Hpo1.

Minor comments:

The second paragraph at the top of page 14 is one sentence and appears to be a statement without context. Was this an error during editing? What this suppose to be the end to the previous paragraph.

**Have all data underlying the figures and results presented in the manuscript been provided?**

Reviewer #1: Yes

Reviewer #2: Yes

PLOS authors have the option to publish the peer review history of their article (what does this mean? ). If published, this will include your full peer review and any attached files.

**Do you want your identity to be public for this peer review?** For information about this choice, including consent withdrawal, please see our Privacy Policy .

Reviewer #1: No

Reviewer #2: No

**Figure resubmission:**
---

## [Editor Report · Decision Letter 1]

19 May 2025

Dear Dr Gaertig,

We are pleased to inform you that your manuscript entitled "Left-right cortical interactions drive intracellular pattern formation  in the ciliate Tetrahymena" has been editorially accepted for publication in PLOS Genetics. Congratulations!

Yours sincerely,

Gregory J. Pazour

Academic Editor

PLOS Genetics

Fengwei Yu

Section Editor

PLOS Genetics

Aimée Dudley

Editor-in-Chief

PLOS Genetics

Anne Goriely

Editor-in-Chief

PLOS Genetics

Comments from the reviewers (if applicable):

**Data Deposition**

http://datadryad.org/submit?journalID=pgenetics&manu=PGENETICS-D-25-00071R1

**Press Queries**

---

## [Editor Report · Acceptance letter]

PGENETICS-D-25-00071R1

Left-right cortical interactions drive intracellular pattern formation  in the ciliate Tetrahymena

Dear Dr Gaertig,

We are pleased to inform you that your manuscript entitled "Left-right cortical interactions drive intracellular pattern formation  in the ciliate Tetrahymena" has been formally accepted for publication in PLOS Genetics! Your manuscript is now with our production department and you will be notified of the publication date in due course.

With kind regards,

Zsofia Freund

PLOS Genetics

On behalf of:
